# Perceptual-GS: Scene-adaptive Perceptual Densification for Gaussian Splatting

**Hongbi Zhou** [1]    **Zhangkai Ni** [1]

## Abstract

3D Gaussian Splatting (3DGS) has emerged as a powerful technique for novel view synthesis. However, existing methods struggle to adaptively optimize the distribution of Gaussian primitives based on scene characteristics, making it challenging to balance reconstruction quality and efficiency. Inspired by human perception, we propose scene-adaptive perceptual densification for Gaussian Splatting (Perceptual-GS), a novel framework that integrates perceptual sensitivity into the 3DGS training process to address this challenge. We first introduce a perception-aware representation that models human visual sensitivity while constraining the number of Gaussian primitives. Building on this foundation, we develop a perceptual sensitivity-adaptive distribution to allocate finer Gaussian granularity to visually critical regions, enhancing reconstruction quality and robustness. Extensive evaluations on multiple datasets, including BungeeNeRF for large-scale scenes, demonstrate that Perceptual-GS achieves state-of-the-art performance in reconstruction quality, efficiency, and robustness. The code is publicly available at: https://github.com/eezkni/Perceptual-GS

## 1. Introduction

Novel view synthesis, which generates images from new viewpoints based on known multi-view images, has been a long-standing focus in computer vision and is further driven by increasing demand from applications such as VR/AR and digital twins. Recently, 3D Gaussian Splatting (3DGS) (Kerbl et al., 2023) has gained significant attention for its exceptional performance, explicitly representing 3D scenes as collections of ellipsoidal Gaussian primitives. Unlike traditional deep learning models, the number of Gaus-

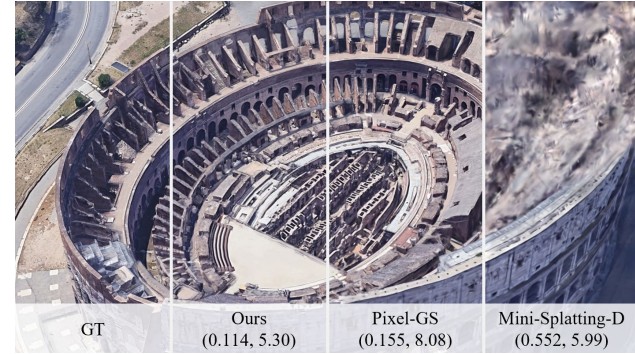

*Figure 1.* The quality-efficiency trade-off and robustness in large-scale scenes of Perceptual-GS are quantified by LPIPS and the number of Gaussians (millions).

sian primitives in 3DGS dynamically evolves during training through adaptive density control, using the average position gradient of Gaussians to determine the need for additional primitives, enhancing the model's capacity to capture fine details in local regions. While this strategy improves overall performance, it struggles with efficiently distributing Gaussians, leading to blurred regions from too few primitives or redundancy from too many.

To enhance the densification capabilities of 3DGS in local regions, numerous studies (Mallick et al., 2024; Xu et al., 2024; Deng et al., 2024; Lyu et al., 2024; Zhang et al., 2025; Liu et al., 2025) have proposed various strategies for improving its performance. Many approaches refine calculating average gradients to more effectively identify Gaussians requiring densification, while others introduce additional metrics for the same purpose. However, these methods often struggle to balance reconstruction quality and computational efficiency, since the number of Gaussian primitives is closely tied to perceptual metrics such as LPIPS (Fang & Wang, 2024). To address this challenge, our research explores whether 3D scenes can be represented with higher perceptual quality using a constrained number of Gaussians. Specifically, we integrate insights from human perception into the training process of 3DGS, adaptively distributing Gaussian primitives according to perceptual sensitivity across different local regions of the scene.

The human visual system (HVS) exhibits characteristics such as contrast sensitivity (Campbell & Robson, 1968), masking effects (Ross & Speed, 1991), and just noticeable

[1]School of Computer Science and Technology, Tongji University, Shanghai, China. Correspondence to: Zhangkai Ni <zkni@tongji.edu.cn>.

*Proceedings of the 42$^{nd}$ International Conference on Machine Learning*, Vancouver, Canada. PMLR 267, 2025. Copyright 2025 by the author(s).

differences (JND) (Shen et al., 2020), which have been extensively utilized in various computer vision tasks. Several studies have integrated certain human perceptual properties with 3DGS (Franke et al., 2025; Lin et al., 2025a), primarily focusing on foveated rendering by leveraging the reduced acuity of HVS in peripheral vision to adjust the precision of different regions dynamically. These methods improve rendering efficiency, but they achieve this by selecting appropriate Gaussian primitives during rendering rather than refining their distribution, inheriting the limitations of vanilla 3DGS. Inspired by the Structural Similarity (SSIM) index (Wang et al., 2004), which suggests that human perception evaluates image quality primarily through local structures (Xue et al., 2014; Ni et al., 2016), we compute gradient magnitude images from various viewpoints of a 3D scene to capture perceptually sensitive local structures. This guides the training process by adaptively distributing Gaussian primitives to represent perceptually sensitive scene details better.

In this paper, we present Perceptual-GS, a novel framework that integrates multi-view perceptual sensitivity into the training process to optimize the distribution of Gaussian primitives. We first enable a perception-aware representation of the scene by precomputing multi-view sensitivity maps using perceptual sensitivity extraction and making each Gaussian perception-aware through dual-branch rendering, constrained by RGB and sensitivity loss during training. Building upon this, we propose a perceptual sensitivity-adaptive distribution, including perceptual sensitivity-guided densification, which enables a sufficient number of Gaussians to represent perceptually critical and poorly learned regions and scene-adaptive depth reinitialization, which further improves performance on scenes with sparse initial point cloud. As shown in Figure 1, Perceptual-GS achieves superior perceptual quality with fewer Gaussians, effectively balancing quality and efficiency. We conduct experiments across multiple datasets, and the results consistently demonstrate a state-of-the-art trade-off between visual fidelity and model complexity, with notable improvements in perceptual metrics and a significant reduction in parameter count even in large-scale scenes. Additionally, Perceptual-GS can be integrated with other 3DGS-based works to enhance performance further, showcasing its generalizability. In summary, our contributions are as follows:

- We design a perception-aware representation that allows each Gaussian primitive to adapt to perceptual sensitivity across different spatial regions efficiently, capturing human perception of the scene in addition to conventional geometry and color.

- We introduce a perceptual sensitivity-adaptive distribution that dynamically allocates Gaussian primitives based on perceptual sensitivity in different areas, achieving a balance between quality and efficiency

while enhancing robustness across diverse scenes.

- Extensive experiments demonstrate that our proposed Perceptual-GS achieves state-of-the-art performance with fewer Gaussian primitives and can be effectively integrated with other 3DGS-based methods, maintaining excellent performance even in large-scale scenes.

## 2. Related Work

### 2.1. 3DGS-based Novel View Synthesis

Novel view synthesis generates images from unseen viewpoints using known views of a 3D scene. Early methods, such as NeRF (Mildenhall et al., 2020) and its variants (Turki et al., 2022; Poole et al., 2023; Ni et al., 2024; Xie et al., 2024), employ neural networks for implicit 3D representation but are constrained by the slow volume rendering. Recently, 3DGS (Kerbl et al., 2023) introduces an efficient method that explicitly represents scenes using ellipsoidal Gaussian primitives for real-time rendering, gaining attention in various applications (Liu et al., 2024; Zhou et al., 2024; Yu et al., 2024b; Lee et al., 2025; Ren et al., 2025). However, 3DGS struggles to distribute the Gaussian primitives efficiently, resulting in redundancy in simple areas and blurriness in texture-rich regions with sparse initial point clouds. Although subsequent quality-focused methods (Fang & Wang, 2024; Zhang et al., 2025) improve the reconstruction quality, they often increase storage requirements and reduce efficiency, while efficiency-focused methods (Lu et al., 2024; Lin et al., 2025a) lower model complexity at the cost of visual fidelity. As a result, balancing quality and efficiency remains a key challenge in 3DGS.

### 2.2. 3DGS Densification

Unlike conventional neural networks with fixed parameter counts, 3DGS initializes Gaussian primitives from point clouds generated through Structure-from-Motion (SfM) and employs adaptive density control to refine local regions. Standard 3DGS uses the average position gradient of Gaussians across all viewpoints to decide primitives to densify, which often leads to blurred details due to insufficient primitives or redundancy from excessive ones. To address this, many methods optimize densification (Du et al., 2024; Li et al., 2024; Yu et al., 2024a; Mallick et al., 2024; Fang & Wang, 2024; Kheradmand et al., 2024; Liu et al., 2025) and most of them refine the metrics used to select Gaussian primitives to be densified. Some enhance the calculation of position gradient with the scaling of Gaussians and frequency domain information (Zhang et al., 2024b; 2025), while others propose additional metrics considering average color gradients and coverage of Gaussian primitives (Kim et al., 2024; Fang & Wang, 2024). However, these methods

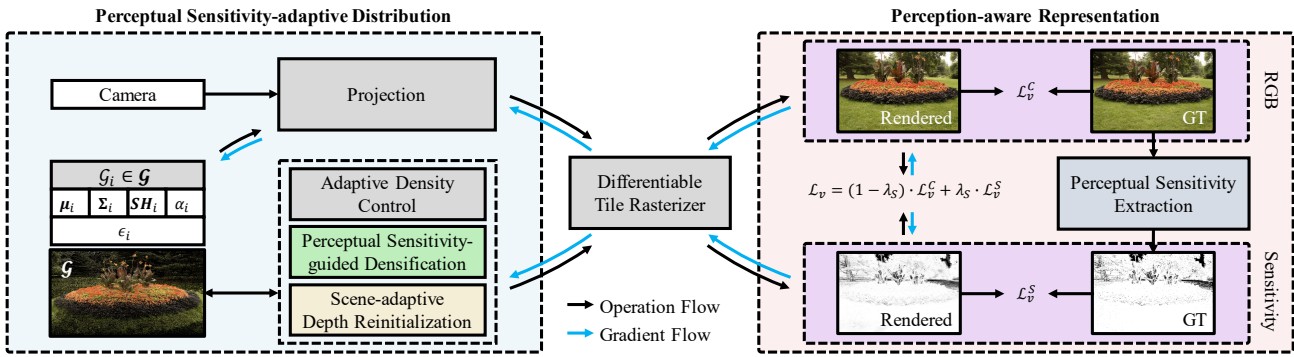

*Figure 2.* Overview of the proposed Perceptual-GS. We first construct a **perception-aware representation** of the scene, enabling each Gaussian primitive to adapt to the perceptual sensitivity of its represented region while constraining the number of Gaussians through perceptual sensitivity extraction and dual-branch rendering. Subsequently, we propose a **perceptual sensitivity-adaptive distribution**, allocating more Gaussians to perceptually critical areas to enhance reconstruction quality and robustness through perceptual sensitivity-guided densification and scene-adaptive depth reinitialization.

often fail to simultaneously consider both visual fidelity and model complexity when selecting Gaussian primitives for densification, making it challenging to achieve a proper balance.

## 3. Methodology

### 3.1. Preliminaries

3DGS renders 2D images from specific viewpoints by projecting 3D Gaussian primitives into 2D space, sorting them according to their distance to the camera, and applying $\alpha$-blending to produce the final image. The set of Gaussian primitives $\boldsymbol{\mathcal{G}}$ is expressed as:

$$\boldsymbol{\mathcal{G}} = \{\mathcal{G}_i(\boldsymbol{\mu}_i, \boldsymbol{\Sigma}_i, \boldsymbol{SH}_i, \alpha_i) \mid i = 1, \dots, N\}, \quad (1)$$

where $\mathcal{G}_i$ is the $i$-th Gaussian primitive, parameterized by its center coordinates $\boldsymbol{\mu}_i$, covariance matrix $\boldsymbol{\Sigma}_i$, opacity $\alpha_i$, and spherical harmonic coefficients $\boldsymbol{SH}_i$ to determine its geometry and color.

Given $\boldsymbol{\mu}_i$ and $\boldsymbol{\Sigma}_i$ of a Gussian primitive $\mathcal{G}_i$, its geometric shape $G_i(\boldsymbol{x})$ can be defined as:

$$G_i(\boldsymbol{x}) = e^{-\frac{1}{2}(\boldsymbol{x}-\boldsymbol{\mu}_i)^\top \boldsymbol{\Sigma}_i^{-1}(\boldsymbol{x}-\boldsymbol{\mu}_i)}, \quad (2)$$

where $\boldsymbol{x}$ is a coordinate in 3D space. The color of a Gaussian primitive $\mathcal{G}_i$ for viewpoint $v$, denoted as $\boldsymbol{C}_i^v$, can be computed using its spherical harmonic coefficients $\boldsymbol{SH}_i$.

After depth-sorting all Gaussian primitives, the rendered RGB color $\mathcal{R}_v^C(\boldsymbol{u})$ at pixel $\boldsymbol{u}$ for viewpoint $v$ is determined by the rendering function:

$$\mathcal{R}_v^C(\boldsymbol{u}) = \sum_{i=1}^{N} \omega_i^v(\boldsymbol{u}) \boldsymbol{C}_i^v, \quad (3)$$

$$\omega_i^v(\boldsymbol{u}) = \alpha_i G_i^v(\boldsymbol{u}) \prod_{j=1}^{i-1} \left(1 - \alpha_j G_j^v(\boldsymbol{u})\right), \quad (4)$$

where $\omega_i^v(\cdot)$ and $G_i^v(\cdot)$ respectively calculate the weight and geometric shape of the elliptical projection of the 3D Gaussian primitive $\mathcal{G}_i$ under viewpoint $v$.

### 3.2. Overview

**Motivation.** In this paper, we aim to enhance the performance of 3DGS while addressing the following challenges:

(a) **Balancing quality and efficiency:** The balance between model quality and efficiency is often neglected when distributing Gaussians, making it challenging to achieve high-fidelity reconstruction without largely increasing rendering overhead, as the number of Gaussians is closely tied to perceptual quality.

(b) **Limited utilization of human perception:** Relying directly on edge maps to assess the perceptually sensitive regions is influenced by response magnitudes, often overlooking subtle structures and reducing accuracy.

(c) **Robustness across different scenes:** Current approaches lack robustness across diverse scenes, particularly in large-scale ones, as they fail to effectively adapt densification to scene-specific properties.

We aim to improve quality and efficiency by prioritizing the densification of Gaussian primitives in high-sensitivity regions to human perception and constraining their generation in low-sensitivity areas, thereby enhancing the perceptual quality of the scene while using fewer Gaussians. Next, we present the framework and detail our four key modules.

**Framework.** Our Perceptual-GS utilizes the high perceptual sensitivity of the human eye to local structures (Xue

et al., 2014) to adaptively identify regions requiring more Gaussian primitives for improved perceptual quality. The pipeline of our proposed method is illustrated in Figure 2, and it can be divided into four individual modules:

(a) **Perceptual Sensitivity Extraction:** Local structures are extracted using traditional edge detection, followed by perception-oriented enhancement and smoothing to generate binary sensitivity maps.

(b) **Dual-branch Rendering:** A novel perceptual sensitivity parameter $\epsilon_i$ is added to each Gaussian primitive in 3D space. Subsequently, the dual-branch rendering strategy is employed to map 2D perceptual sensitivity to 3D primitives while limiting the number of Gaussians in structurally simple regions.

(c) **Perceptual Sensitivity-guided Densification:** Gaussian primitives with high or medium perceptual sensitivity are selectively densified. High-sensitivity regions correspond to areas visually critical to the human eye, while medium-sensitivity regions require more Gaussians for better accuracy.

(d) **Scene-adaptive Depth Reinitialization:** Scenes with sparse initial point cloud derived from Structure-from-Motion (SfM) are identified based on the learning of Gaussian perceptual sensitivity, and depth reinitialization is applied to refine the distribution of Gaussian primitives and enhance reconstruction.

### 3.3. Perceptual Sensitivity Extraction

Image distortion is often assessed by analyzing local structures, as human perception is particularly sensitive to distortions in these areas (Wang et al., 2004). In 3DGS, using local image structures to guide density control is also explored (Mallick et al., 2024; Jiang et al., 2024; Xiang et al., 2024; Lin et al., 2025b). While directly using the derived edge response values can improve reconstruction quality, it has limitations due to large differences in response intensities across various perceptually sensitive regions and may overlook areas that also require densification. For example, in the middle row of Figure 3, the texture of leaves has lower response values compared to the more prominent edges in the first row. These subtle structures, though distinguishable to the human eye, do not promote densification as effectively as the more pronounced edges. To address this, we first capture the human perception of different regions by extracting gradient magnitude maps (Xue et al., 2014) and then apply perception-oriented enhancement to model the thresholding nature of human perception (Lubin, 1997) and perception-oriented smoothing according to the result of eye-tracking studies (Gu et al., 2016). This process retains binary information about pixel perceptibility to the human

eye while discarding absolute response values, as shown in Figure 3.

Specifically, we use the Sobel operator to extract the local structure of the original RGB image $\mathcal{I}$, and the horizontal and vertical gradient convolution kernels $\boldsymbol{G}_x$ and $\boldsymbol{G}_y$ are defined as:

$$\boldsymbol{G}_x = \begin{bmatrix} -1 & 0 & 1 \\ -2 & 0 & 2 \\ -1 & 0 & 1 \end{bmatrix}, \quad \boldsymbol{G}_y = \begin{bmatrix} -1 & -2 & -1 \\ 0 & 0 & 0 \\ 1 & 2 & 1 \end{bmatrix}. \quad (5)$$

Perception-oriented Enhancement

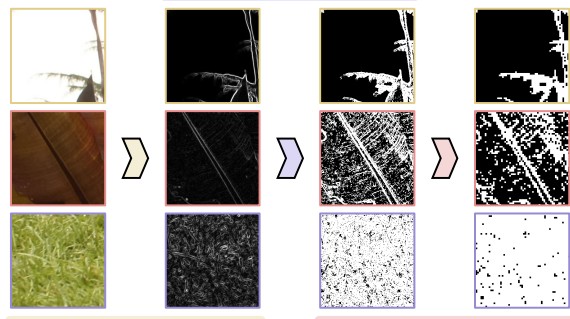

Local Structure Extraction    Perception-oriented Smoothing

*Figure 3.* Pipeline of Perceptual Sensitivity Extraction. An accurate and more prone-to-learn binary sensitivity map that reflects human visual perception can be extracted through this module.

The final edge response map $\boldsymbol{G}$ is computed as:

$$\boldsymbol{G} = \sqrt{(\mathcal{I} \otimes \boldsymbol{G}_x)^2 + (\mathcal{I} \otimes \boldsymbol{G}_y)^2}, \quad (6)$$

where $\otimes$ denotes the convolution operation. After obtaining the gradient magnitude map, we enhance it to better align with human perception. By setting an enhancement threshold $\tau_e$, every pixel value $\boldsymbol{G}(\boldsymbol{u})$ at pixel $\boldsymbol{u}$ in the response map $\boldsymbol{G}$ is binarized to retain only binary information:

$$\boldsymbol{G}_E(\boldsymbol{u}) = \mathbb{I}(\boldsymbol{G}(\boldsymbol{u}) > \tau_e), \quad (7)$$

where $\mathbb{I}(\cdot)$ is the indicator function and $\boldsymbol{G}_E(\boldsymbol{u})$ is the pixel value of the enhanced map at pixel $\boldsymbol{u}$. We further smooth the binary map using average pooling with threshold $\tau_s$, resulting in the final perceptual sensitivity map for the scene.

### 3.4. Dual-branch Rendering

With the 2D perceptual sensitivity maps extracted, mapping them onto 3D Gaussian primitives is essential to make the model perception-aware. A straightforward approach is to accumulate pixel values within the areas covered by the 2D projections of Gaussians from multiple viewpoints (Mallick et al., 2024; Rota Bulò et al., 2025). However, this strategy fails to constrain the sensitivity of different pixels covered

by a single primitive to remain consistent, which undermines the effectiveness of subsequent perceptual sensitivity-guided densification. To efficiently capture human perception from 2D sensitivity maps, we propose a dual-branch rendering framework. In this approach, besides the original RGB branch which renders RGB images of the scene, we introduce a sensitivity branch to render sensitivity maps by associating each Gaussian primitive $\mathcal{G}_i$ with an additional learnable parameter $\epsilon_i$, representing the perceptual sensitivity of Gassians in spatial regions. To ensure consistency and scalability, the sensitivity values are constrained to the range $[0, 1]$ using a sigmoid activation function. This allows sensitivity maps to be rendered similarly to RGB images:

$$\mathcal{R}_v^S(\boldsymbol{u}) = \sum_{i=1}^{N} \omega_i^v(\boldsymbol{u})\sigma(\epsilon_i), \qquad (8)$$

where $\mathcal{R}_v^S(\boldsymbol{u})$ is the value of rendered sensitivity map at pixel $\boldsymbol{u}$ in view $v$, and $\sigma(\cdot)$ represents the sigmoid function.

To optimize the framework, we integrate losses from both branches. For the RGB branch, we follow the loss function $\mathcal{L}_v^C$ as the vanilla 3DGS, which is the weighted sum of L1 and D-SSIM (Wang et al., 2004) loss of view $v$. For the sensitivity branch, the Binary Cross-Entropy (BCE) loss $\mathcal{L}_{BCE}$ is employed to align the rendered sensitivity map $\mathcal{R}_v^S$ with the ground truth $\mathcal{I}_v^S$, and the sensitivity loss $\mathcal{L}_v^S$ of view $v$ is defined as:

$$\mathcal{L}_v^S = \mathcal{L}_{BCE}(\mathcal{I}_v^S, \mathcal{R}_v^S). \qquad (9)$$

The overall loss function $\mathcal{L}_v$ for viewpoint $v$ is defined as a weighted sum of the RGB and sensitivity loss:

$$\mathcal{L}_v = (1 - \lambda_S)\mathcal{L}_v^C + \lambda_S\mathcal{L}_v^S, \qquad (10)$$

where $\lambda_S$ is the weight for the sensitivity loss.

### 3.5. Perceptual Sensitivity-guided Densification

Following the vanilla 3DGS, we initiate the perceptual sensitivity-guided densification after 500 iterations of warm-up, which allows each Gaussian primitive to learn a coarse approximation of geometry, color, and sensitivity, forming a foundation for subsequent densification. To better fit the binarized perceptual sensitivity maps, the well-learned sensitivity of each Gaussian primitive should be close to 0 or 1. Primitives with sensitivity close to 1 are assumed to represent regions with rich local structures, necessitating additional Gaussians for finer detail representation. These Gaussians $\mathcal{G}_h$ are selected using a threshold $\tau_h$:

$$\mathcal{G}_h = \{\mathcal{G}_i \mid \epsilon_i > \tau_h \wedge i \in [1, N]\}. \qquad (11)$$

For Gaussian primitives with significant sensitivity variations across viewpoints, the training process often converges

their sensitivity to incorrect intermediate values to balance discrepancies. These Gaussians need to be split into smaller primitives, as a single primitive cannot adequately capture the complex information within the region. We identify such primitives $\mathcal{G}_m$ using thresholds $\tau_h$ and $\tau_l$:

$$\mathcal{G}_m = \{\mathcal{G}_i \mid \epsilon_i \in [\tau_l, \tau_h] \wedge i \in [1, N]\}. \qquad (12)$$

To prevent excessive densification of Gaussian primitives within objects, we impose weight constraints by a threshold $\tau^\omega$ on selected primitives based on their sensitivity evaluation, and the Gaussian primitives requiring perceptual sensitivity-guided densification $\mathcal{G}_D$ are defined as:

$$\mathcal{G}_D = \{\mathcal{G}_i \mid \omega_i^{max} > \tau^\omega \wedge i \in [1, N]\} \cap (\mathcal{G}_h \cup \mathcal{G}_m), \quad (13)$$

$$\omega_i^{max} = \text{MAX}(\{\sum_{\boldsymbol{u} \in \boldsymbol{pix}_v} \omega_i^v(\boldsymbol{u}) \mid v \in \boldsymbol{V}\}), \qquad (14)$$

where $\text{MAX}(\cdot)$ selects the maximum element in the set, $\boldsymbol{pix}_v$ denotes all pixels in view $v$ and $\omega_i^{max}$ is the maximum weight of Gaussian $\mathcal{G}_i$ across all views $\boldsymbol{V}$ of the scene. This ensures that only the most essential Gaussians are densified.

The vanilla 3DGS employs split and clone operations during densification based on the scaling of Gaussian primitives. Our experiment finds that the split can better capture scene details. Therefore, we only apply the clone operation in perceptual sensitivity-guided densification to $\mathcal{G}_h$ when the scene sensitivity $\beta$ falls below a threshold $\tau_\beta$, indicating scenes with fewer perceptually sensitive regions. Otherwise, we split the selected Gaussians regardless of their scaling. Specifically, the scene sensitivity $\beta$ can be defined as the average pixel sensitivity across all views $\boldsymbol{V}$:

$$\beta = \frac{\sum_{v \in \boldsymbol{V}} avg_v}{|\boldsymbol{V}|}, \qquad (15)$$

$$avg_v = \frac{\sum_{\boldsymbol{u} \in \boldsymbol{pix}_v} v(\boldsymbol{u})}{|\boldsymbol{pix}_v|}, \qquad (16)$$

where $v(\boldsymbol{u})$ is the sensitivity value at pixel $\boldsymbol{u}$ and $avg_v$ denotes the average sensitivity of view $v$.

### 3.6. Scene-adaptive Depth Reinitialization

While perceptual sensitivity-guided densification effectively enhances the perceptual quality of reconstruction, in scenes with sparse initial point clouds, excessive densification of large Gaussians may result in inaccurate distributions. To address this, inspired by (Fang & Wang, 2024), we adaptively apply depth reinitialization on scenes with sparse initial point clouds. The proportion of large Gaussian primitives with medium sensitivity after warm-up $\gamma$ is defined as an indicator of whether the initial point cloud is sparse:

$$\gamma = \frac{|\mathcal{G}_l \cap \mathcal{G}_m|}{|\mathcal{G}_l|}, \qquad (17)$$

$$\mathcal{G}_l = \{\mathcal{G}_i \mid s_i^{max} > Q_3(\mathbf{S}^{max}) \wedge i \in [1, N]\}, \quad (18)$$

where $s_i^{max}$ represents the scaling of the longest axis of $\mathcal{G}_i$, $\mathbf{S}^{max}$ denotes the set of the longest axis scaling of all Gaussians, and $Q_3$ represents the third quartile, identifying the top 25% largest Gaussian primitives $\mathcal{G}_l$. Finally, for scenes where $\gamma$ exceeds a predefined threshold $\tau_\gamma$, depth reinitialization is applied to enhance performance.

### 3.7. Opacity Decline for Clone Operation

In the clone operation of vanilla 3DGS, newly added Gaussian primitives inherit all parameters from the densified primitives, leading to an increase in the opacity of the spatial regions they represent. However, these cloned Gaussian primitives are typically small and insufficiently trained, making them prone to redundancy. As their opacities increase, it becomes more challenging to prune these potentially redundant Gaussians, thereby reducing overall efficiency.

To mitigate the impact of cloning redundant Gaussians on model efficiency, we propose the opacity decline for the clone operation, which reduces the opacity of the spatial regions represented by the cloned Gaussians, thereby facilitating the pruning of redundant Gaussian primitives. According to the alpha-compositing logic, when two Gaussian primitives with an opacity of $\hat{\alpha}$ overlap, the opacity of the corresponding spatial region $A$ can be expressed as:

$$A = \hat{\alpha} + (1 - \hat{\alpha}) \times \hat{\alpha}. \quad (19)$$

Assuming the opacity of the spatial region, *i.e.*, the opacity of the cloned Gaussian primitive, is $\alpha$ before the clone operation, we aim to reduce the opacity $A$ of this region after cloning. Specifically, we apply a transform $\text{OD}(\cdot)$ to $\alpha$ to decline the spatial opacity. The opacity $\hat{\alpha}$ of the two Gaussian primitives after cloning is determined by solving the equation:

$$\hat{\alpha} + (1 - \hat{\alpha}) \times \hat{\alpha} = \text{OD}(\alpha), \quad (20)$$

which yields $\hat{\alpha} = 1 - \sqrt{1 - \text{OD}(\alpha)}$.

When selecting the $\text{OD}(\cdot)$, we aim to apply greater reductions to smaller opacities, encouraging them to be pruned, while applying less reduction to higher opacities to avoid removing important Gaussians. Specifically, we require $\text{OD}(x)$ to be monotonically increasing for $x \in [0, 1]$ and satisfy the following properties:

(a) $\text{OD}(x) \leq x$, indicating that the transformed value is no larger than the original one,

(b) $\text{OD}(0) = 0$, $\text{OD}(1) = 1$, indicating that the transformed value is still in range [0,1],

(c) $a \leq 0.5$, where $a$ represents the unique stationary point of $f(x) = x - \text{OD}(x)$ satisfying its first derivative

$f'(a) = 0$, indicating that smaller opacities are reduced more than larger ones.

In our experiments, we adopt the power function $x^k$ as $\text{OD}(\cdot)$. To determine the optimal value for $k$, we test various exponents and find that larger values of $k$ effectively reduce the number of Gaussian primitives, but may also lead to performance degradation, as shown in Table 1. Although property (c) is not satisfied when $k = 1.0$, we still include this value in the table, indicating that the opacity of the spatial region represented by any primitive remains unchanged after cloning. Ultimately, we select $k = 1.2$ to strike a balance between reconstruction quality and efficiency.

Table 1. The effect of various $k$ values. All metrics are evaluated on the Mip-NeRF 360 dataset and averaged across scenes.

| $k$ | PSNR↑ | SSIM↑ | LPIPS↓ | #G↓ |
|-----|-------|-------|--------|-----|
| 1.0 | 28.02 | 0.839 | 0.172 | 2.74M |
| 1.2 | 28.01 | 0.839 | 0.172 | 2.69M |
| 1.5 | 27.96 | 0.838 | 0.173 | 2.66M |
| 2.0 | 27.99 | 0.838 | 0.174 | 2.60M |

## 4. Experiment

### 4.1. Experiment Setup

**Datasets.** We evaluated the effectiveness of our method across 21 scenes, including 9 scenes from Mip-NeRF 360 (Barron et al., 2022), 2 scenes from Deep Blending (Hedman et al., 2018), 2 scenes from Tanks & Temples (Knapitsch et al., 2017), and 8 scenes from BungeeNeRF (Xiangli et al., 2022).

**Baselines.** We select quality-focused related works that enhance 3DGS performance through optimizing densification strategies, similar to Perceptual-GS, for comparison to validate the effectiveness of our proposed method. Specifically, we chose state-of-the-art methods, including Pixel-GS (Zhang et al., 2025), Mini-Splatting-D (Fang & Wang, 2024), Taming-3DGS (Mallick et al., 2024), and the vanilla 3DGS (Kerbl et al., 2023) as baselines.

Since 3DGS and Pixel-GS did not provide metrics for the number of Gaussian primitives in their original papers, we retrain both models to obtain these values, denoted as 3DGS* and Pixel-GS*. Additionally, to ensure a fair comparison of FPS and avoid discrepancies caused by testing on different devices, we re-evaluated the rendering speed of various methods. As none of the baselines report results on BungeeNeRF, although it is commonly used to evaluate other 3DGS-based methods (Lu et al., 2024; Ren et al., 2025; Chen et al., 2025), we retrain all models on this dataset and use the data from the original papers for all other metrics.

**Implementation Details.** We align the experimental setup

*Table 2.* Quantitative results on reconstruction quality, comparing our method with state-of-the-art methods in terms of PSNR↑, SSIM↑ and LPIPS↓. The best , second-best , and third-best results are highlighted.

| Method | Mip-NeRF 360 | | | Tanks & Temples | | | Deep Blending | | | BungeeNeRF | | |
|---|---|---|---|---|---|---|---|---|---|---|---|---|
| | PSNR↑ | SSIM↑ | LPIPS↓ | PSNR↑ | SSIM↑ | LPIPS↓ | PSNR↑ | SSIM↑ | LPIPS↓ | PSNR↑ | SSIM↑ | LPIPS↓ |
| 3DGS* | 27.71 | 0.826 | 0.202 | 23.61 | 0.845 | 0.178 | 29.54 | 0.900 | 0.247 | 27.64 | 0.912 | 0.100 |
| Pixel-GS* | 27.85 | 0.834 | 0.176 | 23.71 | 0.853 | 0.152 | 28.92 | 0.893 | 0.250 | OOM in 1 scene | | |
| Mini-Splatting-D | 27.51 | 0.831 | 0.176 | 23.23 | 0.853 | 0.140 | 29.88 | 0.906 | 0.211 | 25.58 | 0.861 | 0.149 |
| Taming-3DGS | 27.79 | 0.822 | 0.205 | 24.04 | 0.851 | 0.170 | 30.14 | 0.907 | 0.235 | OOM in 2 scenes | | |
| Ours | 28.01 | 0.839 | 0.172 | 23.90 | 0.857 | 0.151 | 29.94 | 0.907 | 0.231 | 27.86 | 0.918 | 0.095 |

Ground Truth  Ours  Taming-3DGS

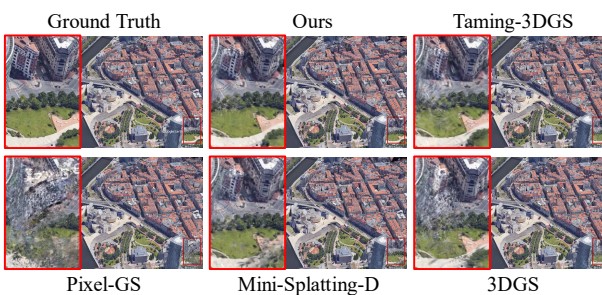

Pixel-GS  Mini-Splatting-D  3DGS

*Figure 4.* A qualitative comparison of Bilbao in BungeeNeRF.

*Table 3.* Quantitative results on reconstruction efficiency, comparing our method with state-of-the-art methods in terms of the number of Gaussian primitives (#G)↓ and rendering speed (FPS)↑.

| Method | Mip-NeRF 360 | | Tanks & Temples | | Deep Blending | | BungeeNeRF | |
|---|---|---|---|---|---|---|---|---|
| | #G↓ | FPS↑ | #G↓ | FPS↑ | #G↓ | FPS↑ | #G↓ | FPS↑ |
| 3DGS* | 3.14M | 193 | 1.83M | 247 | 2.81M | 194 | 6.92M | 69 |
| Pixel-GS* | 5.23M | 105 | 4.49M | 101 | 4.63M | 114 | OOM in 1 scene | |
| Mini-Splatting-D | 4.69M | 120 | 4.28M | 115 | 4.63M | 159 | 6.08M | 86 |
| Taming-3DGS | 3.31M | 122 | 1.84M | 149 | 2.81M | 130 | OOM in 2 scenes | |
| Ours | 2.69M | 166 | 1.72M | 218 | 2.86M | 178 | 4.97M | 89 |

with the baselines, and the settings for the newly introduced hyperparameters in Perceptual-GS are provided in the Appendix. To achieve a better balance between quality and efficiency, we use different weight thresholds $\tau^\omega$ for high- and medium-sensitivity Gaussians, denoted as $\tau_h^\omega$ and $\tau_m^\omega$, respectively. All training and testing are conducted on a single NVIDIA RTX4090 GPU with 24GB of memory.

**Metrics.** To evaluate the performance of different methods, we use common metrics including PSNR, SSIM (Wang et al., 2004), and LPIPS (Zhang et al., 2018). Besides, we consider the number of Gaussian primitives (#G) in millions (M) and rendering speed (FPS). These metrics highlight the superior trade-off between quality and efficiency achieved by our approach.

### 4.2. Comparisons with State-of-the-art

**Quantitative Comparison.** Table 2 shows the quantitative comparison of Perceptual-GS with state-of-the-art methods in novel view synthesis on reconstruction quality. Across four datasets, our proposed Perceptual-GS achieves superior reconstruction quality, particularly excelling in SSIM and the perceptually relevant LPIPS metric. Unlike Pixel-GS and Taming-3DGS which face CUDA out-of-memory (OOM) issues due to excessive Gaussians in large-scale scenes, Perceptual-GS adaptively distributes primitives based on the perceptual sensitivity of different regions, achieving a superior quality-efficiency trade-off.

**Qualitative Comparison.** The proposed Perceptual-GS allocates more Gaussians to object details and edges, ef-

fectively reducing scene blurriness. As shown in Figure 4, our proposed Perceptual-GS accurately reconstructs roads, buildings, and grassland at scene boundaries, avoiding artifacts seen in other methods. Similarly, in Figure 5, our method captures ground textures more faithfully, while other methods tend to produce more artifacts in these regions.

**Efficiency Comparison.** Table 3 provides a quantitative analysis of model complexity and rendering efficiency. Comparing with other quality-focused methods, Perceptual-GS demonstrates a significant improvement in efficiency and the quality-efficiency balance, rendering high-fidelity novel views with faster speed and fewer Gaussian primitives.

### 4.3. Ablation Study

**Effectiveness of Perceptual Sensitivity Extraction.** We evaluate the impact of enhanced sensitivity maps for guiding densification by comparing them to edge response maps derived from the Sobel operator. Since scene-adaptive depth reinitialization does not function properly without perception-oriented enhancement, we exclude depth reinitialization for all scenes during the experiments and apply only split in perceptual sensitivity-guided densification. As shown in the "w/o PE" results in Table 4, the synthesized novel views exhibit similar reconstruction quality to vanilla 3DGS without perception-oriented enhancement (PE) since the inaccurate learning of sensitivity.

**Effectiveness of Perceptual Sensitivity-guided Densification.** Perceptual sensitivity-guided densification is a key component of our method. To assess the individual contribu-

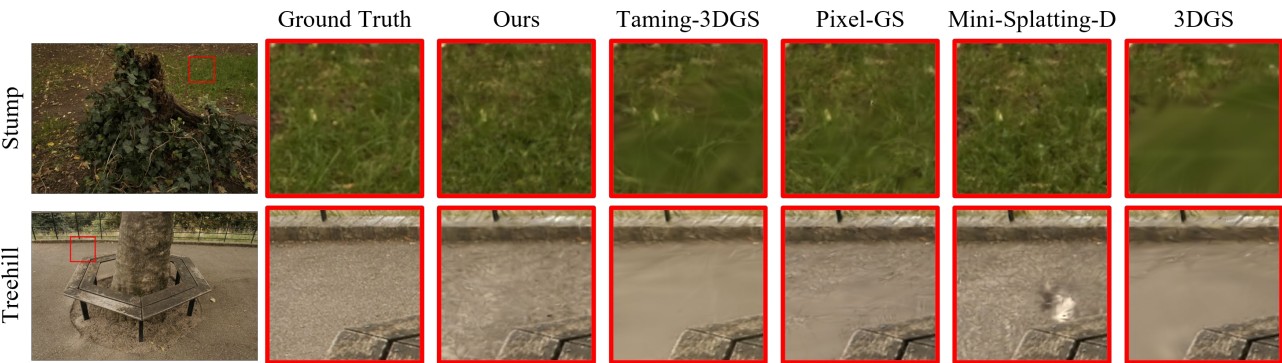

*Figure 5.* A qualitative comparison of Perceptual-GS with other methods on Stump and Treehill in Mip-NeRF 360.

*Table 4.* Ablation studies on various modules of Perceptual-GS. All metrics are evaluated on the Mip-NeRF 360 dataset and averaged across all scenes.

| | PSNR↑ | SSIM↑ | LPIPS↓ | #G↓ |
|---|---|---|---|---|
| FULL | 28.01 | 0.839 | 0.172 | 2.69M |
| 3DGS* | 27.71 | 0.826 | 0.202 | 3.14M |
| w/o PE | 27.74 | 0.825 | 0.204 | 2.09M |
| w/o HD | 27.74 | 0.826 | 0.204 | 2.02M |
| w/o MD | 27.86 | 0.831 | 0.179 | 2.56M |
| w/o SDR | 27.93 | 0.832 | 0.176 | 2.68M |
| w/o OD | 27.99 | 0.839 | 0.172 | 3.25M |

tions of densifying high- and medium-sensitivity Gaussians, we conduct ablation studies by separately removing their densification processes. The results, presented as "w/o HD" in Table 4, show that excluding high-sensitivity Gaussian densification (HD) reduces Gaussian primitives significantly, causing the performance to degrade to levels comparable to vanilla 3DGS. Similarly, removing medium-sensitivity Gaussian densification (MD) impairs the accurate reconstruction of detailed regions. However, as shown in "w/o MD" in Table 4, it still achieves notable improvements over vanilla 3DGS thanks to the dual-branch rendering which reduces the proportion of medium-sensitivity Gaussians.

**Effectiveness of Scene-adaptive Depth Reinitialization.** To verify that the improvement of the Perceptual-GS is not solely attributed to depth reinitialization, we perform an ablation study on the scene-adaptive depth reinitialization (SDR). The results in "w/o SDR" in Table 4 show that our method achieves improved outcomes even without depth reinitialization, maintaining a balance between quality and efficiency to achieve state-of-the-art performance.

**Effectiveness of Opacity Decline.** Our proposed Opacity Decline (OD) mechanism for clone operation in densification encourages the removal of redundant Gaussian primitives while preserving similar visual quality. As shown in Table 4, with Opacity Decline applied, Perceptual-GS achieves comparable performance in quality metrics using significantly fewer Gaussians, demonstrating its effective-

ness in removing redundant primitives.

### 4.4. Analysis

**Integrating with existing works.** In Table 5 and Table 6, we integrate our proposed framework with vanilla 3DGS and Pixel-GS, denoted as w/ Ours, further demonstrating its effectiveness. The proposed method achieves significant improvements across all quality metrics on both baselines, while also reducing the number of Gaussian primitives in most datasets, thereby enhancing efficiency.

It is worth noting that our method remains effective even under sparse-view settings. In Table 7, we integrate the proposed method with CoR-GS (Zhang et al., 2024a) and conduct quantitative comparison on the 24-view Mip-NeRF 360 dataset, which also demonstrates a significant performance improvement. Since the original paper did not provide the 24-view dataset, we retrain the model, denoted as CoR-GS*, using a dataset reconstructed according to the instructions in the official released code. The versatility of our method enables its integration with other approaches to achieve even better performance.

**Effectiveness of dual-branch rendering.** In addition to mapping perceptual sensitivity, our experiments reveal that dual-branch rendering (DBR) also reduces the number of Gaussian primitives. As shown in Table 8, we compare the performance and efficiency of the vanilla 3DGS, 3DGS with OD, and with both DBR and OD to demonstrate its effect. The results indicate that DBR can slightly constrain the number of Gaussians while maintaining comparable quality since low-sensitivity Gaussians with well-learned sensitivity exhibit lower sensitivity loss. After weighting, their total loss is reduced, preventing them from reaching the position gradient threshold and thereby suppressing densification.

**Rendering quality in low-sensitive regions.** Although the dual-branch rendering strategy suppresses the densification of Gaussian primitives in low-sensitivity regions, it does not compromise the reconstruction quality in these areas. In Table 9, we use the perceptual sensitivity map as a mask

*Table 5.* The quantitative result of the proposed method is based on different models on Mip-NeRF 360, Tanks & Temples, and Deep Blending. Metrics are averaged across the scenes. The improvements and reductions in the metrics are highlighted.

| Method | Mip-NeRF 360 | | | | Tanks & Temples | | | | Deep Blending | | | |
|---|---|---|---|---|---|---|---|---|---|---|---|---|
| | PSNR↑ | SSIM↑ | LPIPS↓ | #G↓ | PSNR↑ | SSIM↑ | LPIPS↓ | #G↓ | PSNR↑ | SSIM↑ | LPIPS↓ | #G↓ |
| 3DGS* | 27.71 | 0.826 | 0.202 | 3.14M | 23.61 | 0.845 | 0.178 | 1.83M | 29.54 | 0.900 | 0.247 | 2.81M |
| w/ Ours | 28.01 | 0.839 | 0.172 | 2.69M | 23.90 | 0.857 | 0.151 | 1.72M | 29.94 | 0.907 | 0.231 | 2.86M |
| Δ | +0.30 | +0.013 | -0.030 | -0.45M | +0.29 | +0.012 | -0.027 | -0.11M | +0.40 | +0.007 | -0.016 | +0.05M |
| Pixel-GS* | 27.85 | 0.834 | 0.176 | 5.23M | 23.71 | 0.853 | 0.152 | 4.49M | 28.92 | 0.893 | 0.250 | 4.63M |
| w/ Ours | 28.01 | 0.841 | 0.167 | 3.37M | 23.95 | 0.859 | 0.142 | 2.96M | 29.71 | 0.901 | 0.233 | 3.59M |
| Δ | +0.16 | +0.007 | -0.009 | -1.86M | +0.24 | +0.006 | -0.010 | -1.53M | +0.79 | +0.008 | -0.017 | -1.04M |

*Table 6.* The quantitative result of the proposed method is based on different models on BungeeNeRF. We present metrics averaged on the dataset and from three single scenes.

| Method | BungeeNeRF | | | | Pompidou | | | | Chicago | | | | Amsterdam | | | |
|---|---|---|---|---|---|---|---|---|---|---|---|---|---|---|---|---|
| | PSNR↑ | SSIM↑ | LPIPS↓ | #G↓ | PSNR↑ | SSIM↑ | LPIPS↓ | #G↓ | PSNR↑ | SSIM↑ | LPIPS↓ | #G↓ | PSNR↑ | SSIM↑ | LPIPS↓ | #G↓ |
| 3DGS* | 27.64 | 0.912 | 0.100 | 6.92M | 27.00 | 0.916 | 0.095 | 9.11M | 27.97 | 0.927 | 0.086 | 6.32M | 27.60 | 0.913 | 0.100 | 6.19M |
| w/ Ours | 27.86 | 0.918 | 0.095 | 4.97M | 27.18 | 0.922 | 0.089 | 6.12M | 28.39 | 0.933 | 0.081 | 4.48M | 27.89 | 0.922 | 0.087 | 4.96M |
| Δ | +0.22 | +0.006 | -0.005 | -1.95M | +0.18 | +0.006 | -0.006 | -2.99M | +0.42 | +0.006 | -0.005 | -1.84M | +0.29 | +0.009 | -0.013 | -1.23M |
| Pixel-GS* | OOM in 1 scene | | | | OOM | | | | 27.52 | 0.921 | 0.090 | 9.76M | 27.76 | 0.916 | 0.095 | 10.26M |
| w/ Ours | 27.64 | 0.913 | 0.100 | 5.92M | 27.01 | 0.918 | 0.092 | 7.39M | 28.36 | 0.930 | 0.081 | 5.58M | 27.98 | 0.922 | 0.085 | 6.60M |
| Δ | — | — | — | — | — | — | — | — | +0.84 | +0.009 | -0.009 | -4.18M | +0.22 | +0.006 | -0.010 | -3.66M |

*Table 7.* The quantitative result of the proposed method is based on CoR-GS on 24-view Mip-NeRF 360. Metrics are averaged across the scenes.

| | PSNR↑ | SSIM↑ | LPIPS↓ |
|---|---|---|---|
| CoR-GS* | 22.26 | 0.664 | 0.341 |
| w/Ours | 22.42 | 0.681 | 0.281 |
| Δ | +0.16 | +0.017 | -0.060 |

*Table 8.* Effect of dual-branch rendering on constraining the number of Gaussians.

| | PSNR↑ | SSIM↑ | LPIPS↓ | #G↓ |
|---|---|---|---|---|
| 3DGS* | 27.71 | 0.826 | 0.202 | 3.14M |
| +OD | 27.74 | 0.825 | 0.207 | 2.22M |
| +OD +DBR | 27.69 | 0.822 | 0.212 | 1.94M |

to retain only the low-sensitive pixels and compare our method with the vanilla 3DGS. The results validate that Perceptual-GS achieves comparable rendering performance in low-sensitivity regions.

**Effectiveness on large-scale scenes.** As shown in Figure 1, Table 2 and Table 3, our method demonstrates great robustness in large-scale scenes, avoiding introducing excessive Gaussians like Pixel-GS and the reconstruction failures in Mini-Splatting-D. This is largely attributed to our dual-branch rendering and perceptual sensitivity-guided densification, which limit the number of densified Gaussians while adaptively identifying regions requiring more primitives.

*Table 9.* Quantitative comparison between Perceptual-GS and the vanilla 3DGS on rendering results masked by the perceptual sensitivity map.

| | PSNR↑ | SSIM↑ | LPIPS↓ |
|---|---|---|---|
| 3DGS* | 40.28 | 0.990 | 0.014 |
| Ours | 40.72 | 0.991 | 0.014 |

## 5. Conclusion

In this paper, we introduce Perceptual-GS, leveraging scene-adaptive perceptual densification to achieve superior perceptual quality with constrained Gaussian primitives. The proposed method promotes the densification of Gaussians in high-sensitivity regions according to human perception while suppressing it in low-sensitivity areas, achieving a balance between reconstruction quality and efficiency. Specifically, we first extract local scene structures using gradient magnitude maps and enhance them based on the characteristics of human perception. During training, a dual-branch rendering strategy maps 2D sensitivity onto 3D Gaussians and constrains the number of primitives. In addition to the adaptive density control in vanilla 3DGS, we densify high- and medium-sensitivity Gaussians to improve reconstruction quality. Finally, scene-adaptive depth reinitialization is applied for better performance. Extensive experiments on multiple datasets demonstrate that our method effectively balances reconstruction quality and efficiency, achieving state-of-the-art performance in novel view synthesis tasks.

## Acknowledgement

This work was supported in part by the National Natural Science Foundation of China under Grant 62201387 and in part by the Fundamental Research Funds for the Central Universities.

## Impact Statement

This paper focuses on advancing 3DGS-based Novel View Synthesis, guiding the training process of 3DGS with human perception for a better trade-off between quality and efficiency. While our work may have potential societal implications, none require specific emphasis at this time.

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

# Appendix

## A. Implementation Details

We adopt the default settings of 3DGS and show the additional hyperparameters introduced in Perceptual-GS in Table 10.

*Table 10.* Definition and value of hyperparameters introduced in Perceptual-GS.

| $H.P.$ | Definition | value |
|---|---|---|
| $\tau_e$ | perception-oriented enhancement threshold | 0.05 |
| $\tau_s$ | perception-oriented smoothing threshold | 0.3 |
| $\lambda_S$ | sensitivity loss weight | 0.1 |
| $Iter_h$ | high-sensitivity Gaussians densification interval | 1000 |
| $Iter_m$ | medium-sensitivity Gaussians densification interval | 1500 |
| $\tau_h$ | high-sensitivity Gaussians threshold of perceptual sensitivity | 0.9 |
| $\tau_l$ | low-sensitivity Gaussians threshold of perceptual sensitivity | 0.3 |
| $\tau_h^{\omega}$ | high-sensitivity Gaussians threshold of weight | 25 |
| $\tau_m^{\omega}$ | medium-sensitivity Gaussians threshold of weight | 10 |
| $\tau_\beta$ | high-sensitivity scenes threshold | 0.85 |
| $\tau_\gamma$ | scenes with sparse initial point cloud threshold | 0.55 |

## B. Additional Qualitative Comparisons with State-of-the-art

We provide additional qualitative comparisons in this section to further showcase the superior visual quality, efficiency, and balance achieved by Perceptual-GS. As shown in Figure 6, the proposed method excels in reconstructing intricate details, such as the complete shadow on the crosswalk in Amsterdam. Figure 7 demonstrates that our method generates novel views with fewer blurred regions while achieving better efficiency in both storage and rendering speed. Besides, our method achieves better performance in depth rendering, as illustrated in Figure 8. Compared to Pixel-GS, the proposed method reconstructs scene geometry with higher accuracy. These results underline the robustness and effectiveness of Perceptual-GS, particularly in large-scale scenes.

We also compare the qualitative results of integrating our proposed method with different existing approaches, as shown in Figure 9, Figure 10, and Figure 11, where our method is respectively integrated with the vanilla 3DGS, the quality-focused Pixel-GS, and CoR-GS designed for sparse-view settings, denoted as w/ Ours. Our method significantly reduces blurriness in the scenes and is able to reconstruct some texture details more clearly.

To better demonstrate the effect of the perceptual sensitivity map, we present the distribution of Gaussian primitives in regions with different sensitivity in Figure 12 and rendered sensitivity maps during training in Figure 13. Furthermore, Figure 14 shows the rendered results with the perceptually sensitive regions masked and compares our method with the vanilla 3DGS. The results indicate that there is no compromise in reconstruction quality in low-sensitive regions, even though our dual-branch rendering strategy reduces the number of primitives in these areas.

## C. Visualization of Ablation Studies

To visually demonstrate the impact of each module on Perceptual-GS, we provide the visualization of our ablation studies. As shown in "w/o PE", "w/o HD" and "w/o MD" in Figure 15, compared with the full model, they exhibit more blurriness in detailed areas, with a noticeable decline in reconstruction quality. In contrast, without scene-adaptive depth reinitialization and opacity decline, the model still maintains a similar visual effect to the full one, demonstrating the effectiveness of our proposed perceptual sensitivity-guided densification, as illustrated in "w/o SDR" and "w/o OD" in Figure 15.

## D. Ablation on Hyperparameters

We conduct ablation studies on several key hyperparameters to assess their impact on our proposed method.

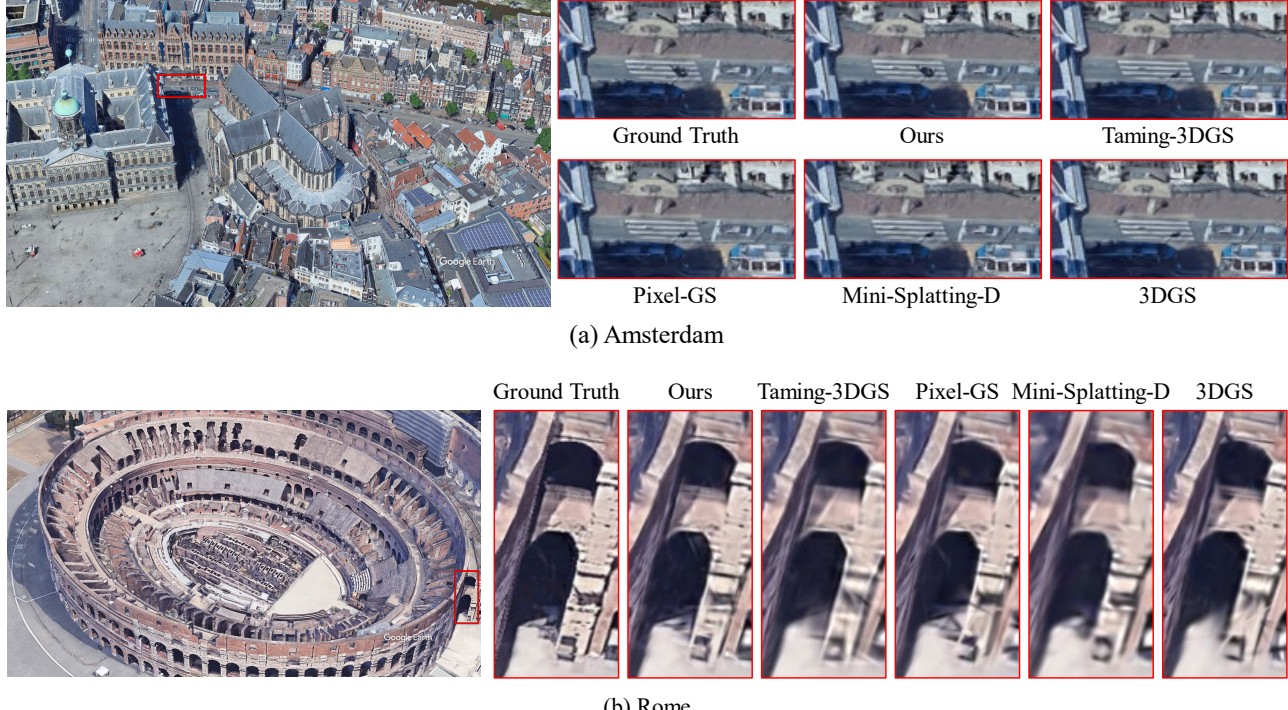

Figure 6. A qualitative comparison of Perceptual-GS with other methods on Amsterdam and Rome in BungeeNeRF.

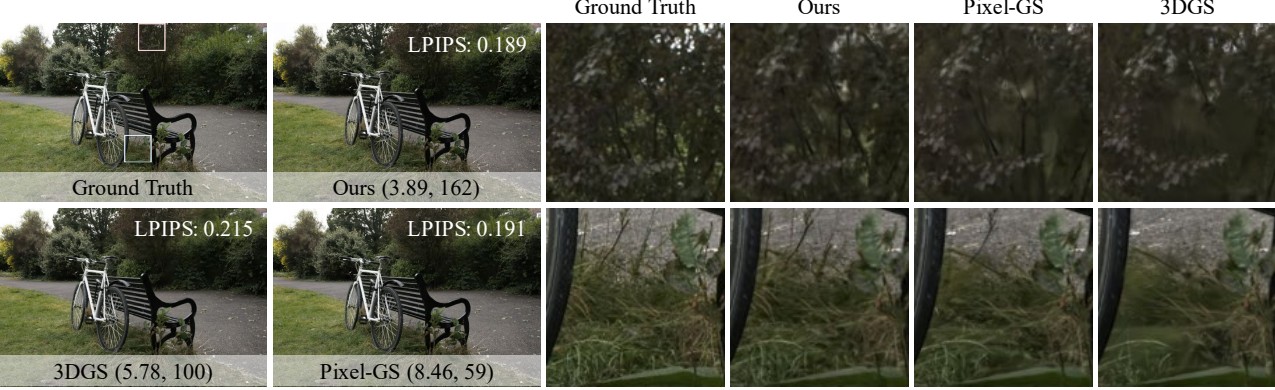

Figure 7. Qualitative efficiency results on Bicycle in Mip-NeRF 360 show that our approach achieves superior visual quality compared to the quality-focused method Pixel-GS, using less than half the number of Gaussian primitives and more than doubling the rendering speed. The number of Gaussians (in millions) and FPS are shown as (Number, FPS).

(a) **Weight of Sensitivity Loss** $\lambda_S$: The balance between the contributions of the two rendering branches to the final optimization can be adjusted by modifying $\lambda_S$. As shown in Table 11, increasing the weight of the sensitivity branch effectively reduces the number of Gaussian primitives while maintaining relatively high perceptual quality comparing with the vanilla 3DGS. However, for better reconstruction quality, we adopt a lower value for $\lambda_S$.

(b) **Threshold of Weight** $\tau_h^\omega$ **and** $\tau_m^\omega$: We evaluate the performance with different values of the weight thresholds $\tau_h^\omega$ and $\tau_m^\omega$ in Table 11. Since high-sensitivity Gaussian primitives represent more complex structures, a lower threshold increases their densification. As the threshold decreases, perceptual quality improves slightly, but more Gaussian primitives are introduced. Similarly, $\tau_m^\omega$ affects quality and efficiency, though to a lesser extent, because dual-branch rendering drives the sensitivity of more Gaussian primitives toward 0 or 1, resulting in fewer medium-sensitivity Gaussians. Therefore, we select a relatively higher value for $\tau_h^\omega$ to achieve a better trade-off and a lower value for $\tau_m^\omega$

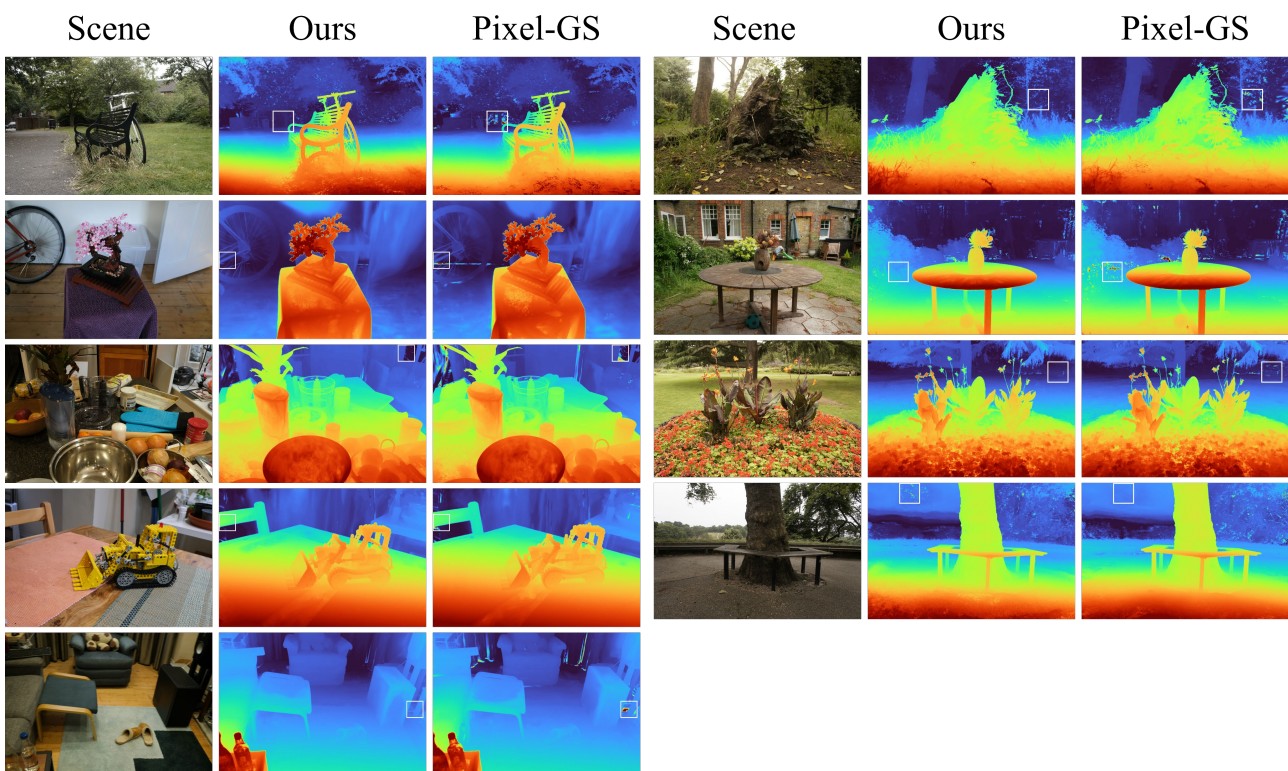

*Figure 8.* A qualitative comparison of the rendering depth between Perceptual-GS and Pixel-GS on Mip-NeRF 360.

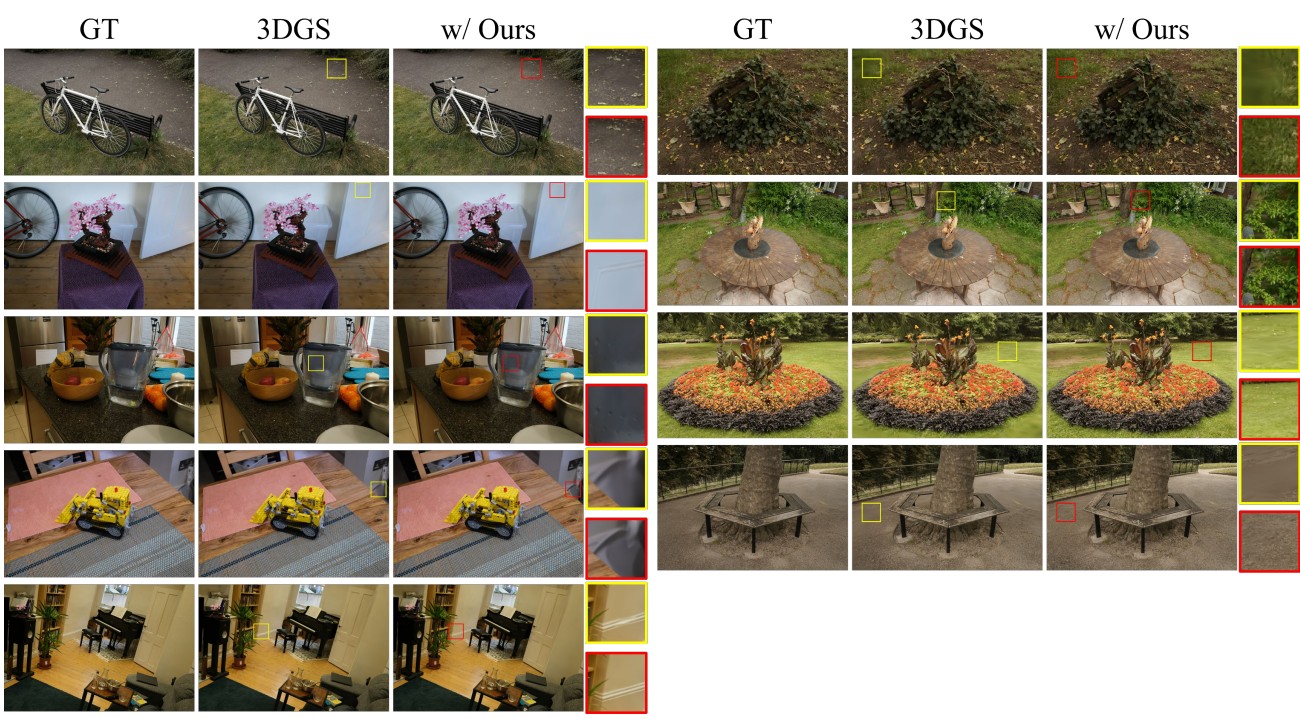

*Figure 9.* The qualitative result of the proposed method is based on the vanilla 3DGS on Mip-NeRF 360.

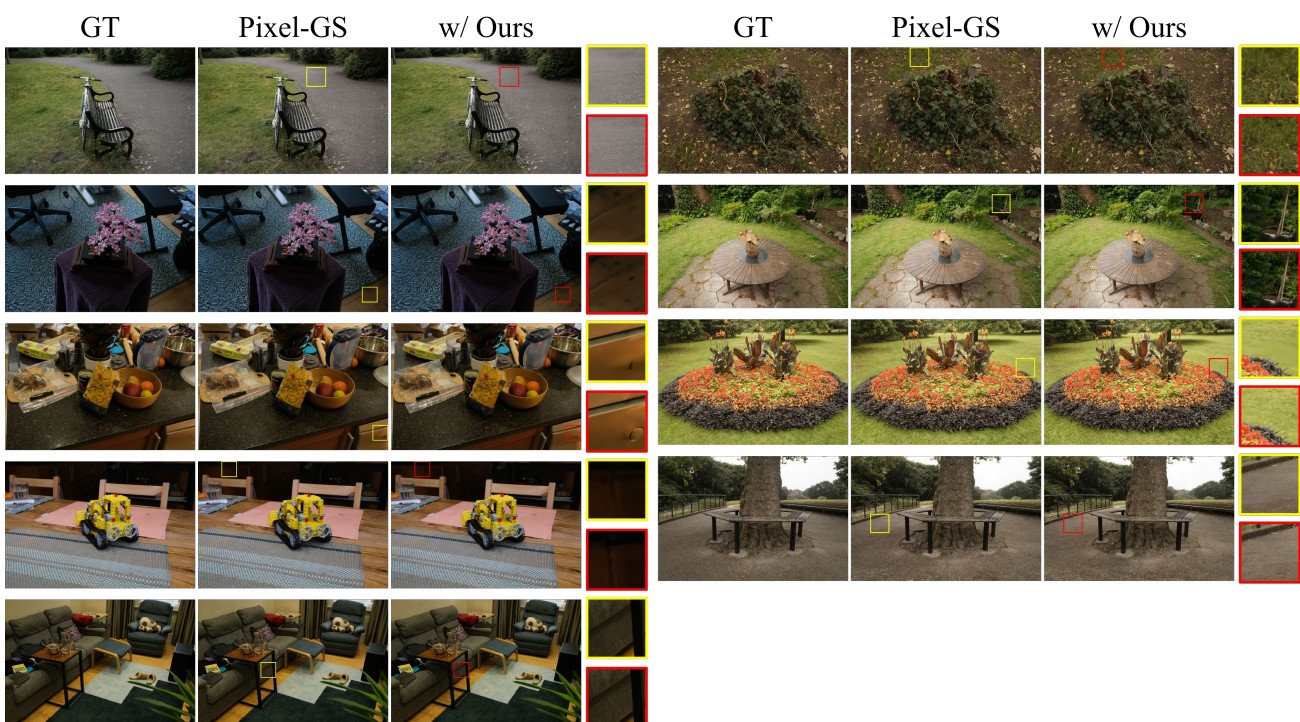

*Figure 10.* The qualitative result of the proposed method is based on Pixel-GS on Mip-NeRF 360.

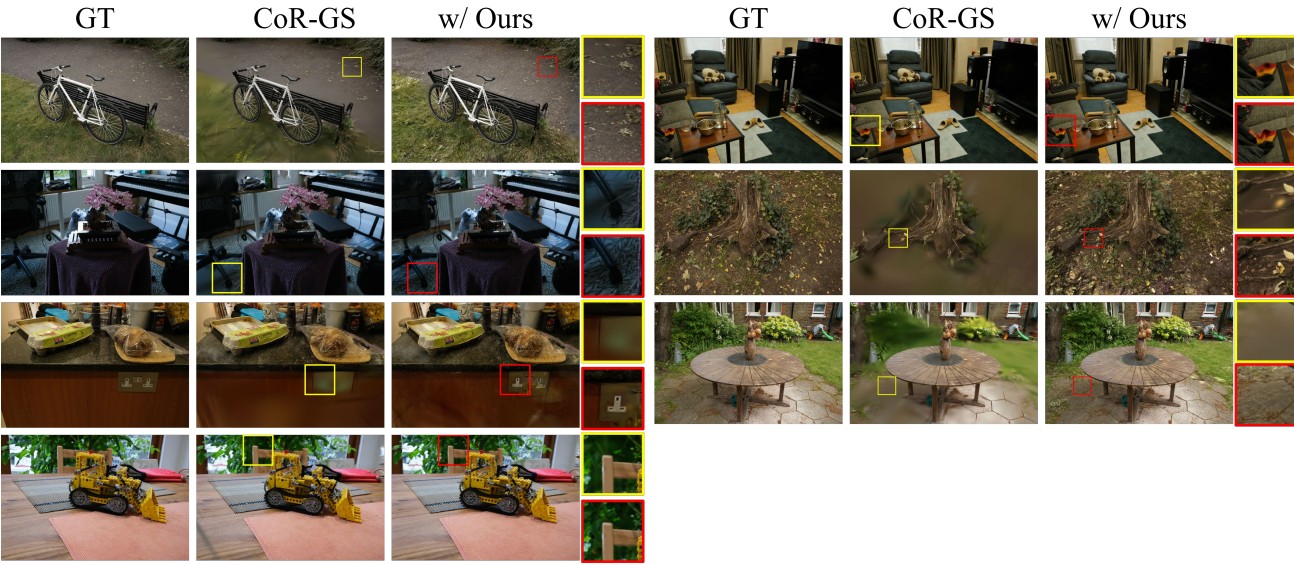

*Figure 11.* The qualitative result of the proposed method is based on CoR-GS on 24-view Mip-NeRF 360.

to prioritize quality.

(c) **Densification Interval** $Iter_h$ **and** $Iter_m$**:** To determine the optimal densification intervals, we experiment with different values of $Iter_h$ and $Iter_m$, as shown in Table 11. The densification intervals for high- and medium-sensitivity Gaussians, like $\tau_h^\omega$ and $\tau_m^\omega$, also influence the model's quality and efficiency. We find that their effects are similar, so we use a smaller $Iter_h$ to improve reconstruction quality while selecting a slightly larger $Iter_m$ to identify medium-sensitivity Gaussian primitives during optimization better.

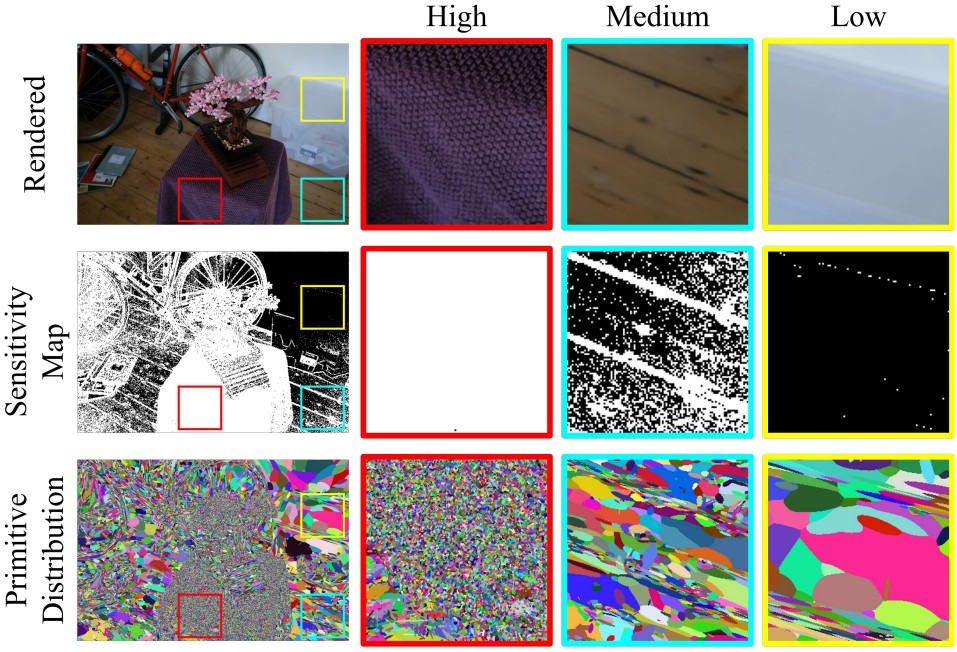

*Figure 12.* The visualization of the effect of perceptual sensitivity map in different spatial regions. Perceptual-GS distributes more primitives to perceptually sensitive regions.

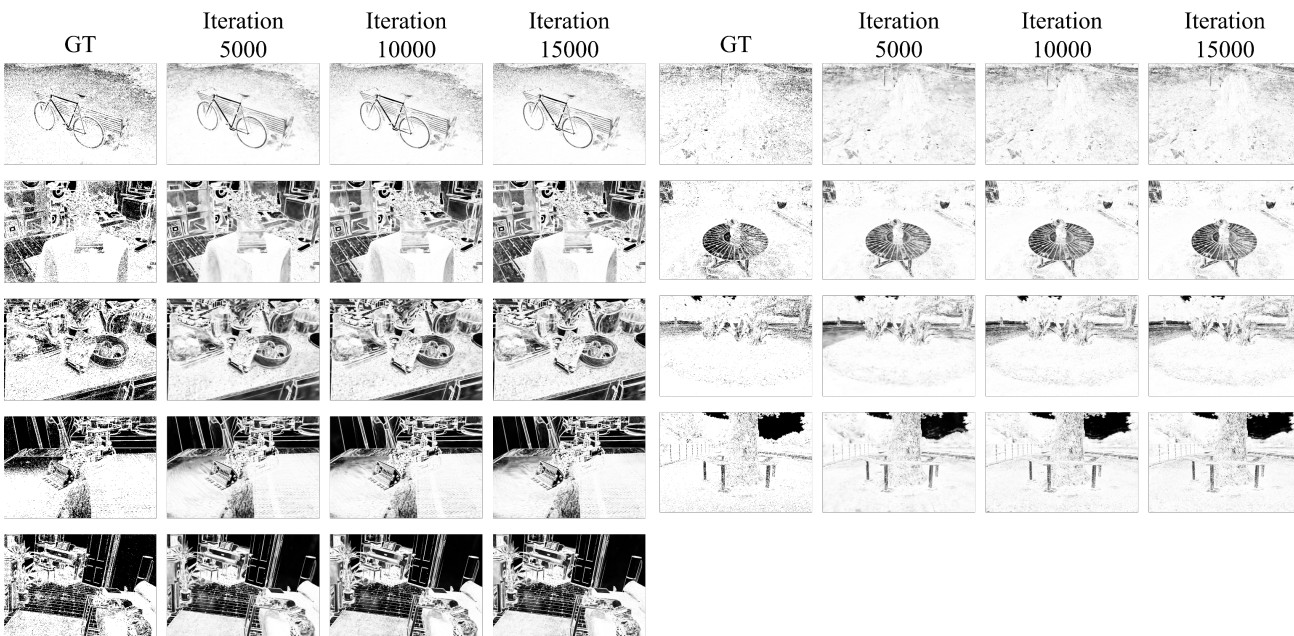

*Figure 13.* The visualization of perceptual sensitivity maps rendered during the training process.

## E. Per Scene Quantitative Comparisons with State-of-the-art

We present per-scene quantitative comparisons with existing methods to further illustrate the improvements in quality, efficiency, and their balance achieved by Perceptual-GS, as shown in Table 12, Table 13, Table 14, Table 15, Table 16, Table 17, Table 18, Table 19, Table 20, Table 21, Table 22, Table 23. To evaluate the overall performance of the model in

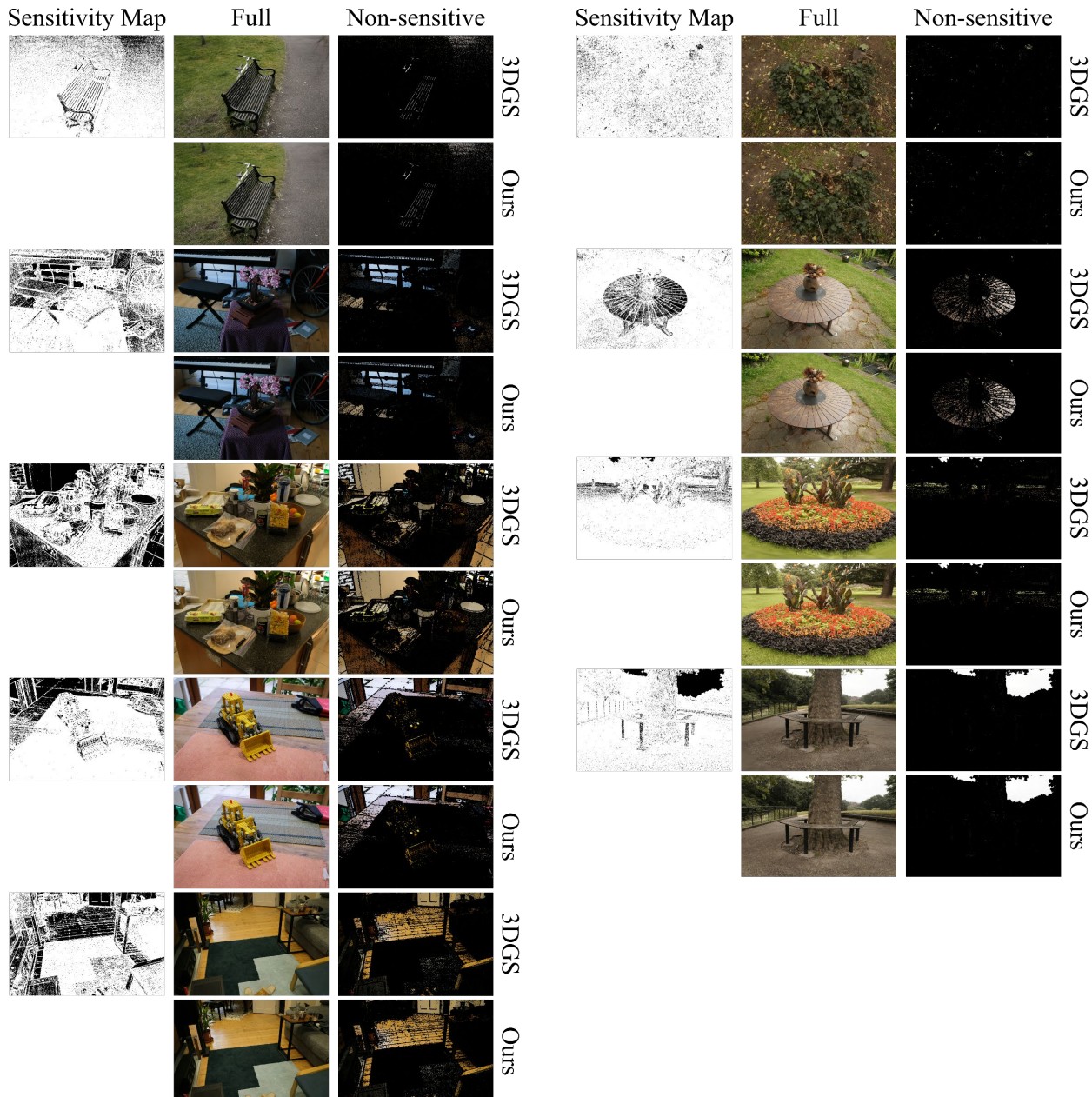

*Figure 14.* Qualitative comparison of the reconstruction quality of low-sensitive regions between 3DGS and the proposed method.

terms of both efficiency and perceptual quality, we introduce a new metric, QEB:

$$QEB = \frac{100 \times \#G \times \text{LPIPS}}{\text{FPS}}, \tag{21}$$

which jointly considers rendering quality and efficiency, and serves as a reference for balancing the trade-off between reconstruction fidelity and speed. The proposed method demonstrates notable improvements in perceptual metrics such as LPIPS, along with a significant reduction in the number of Gaussian primitives. Notably, Perceptual-GS achieves a superior quality-efficiency trade-off in large-scale scenes from BungeeNeRF, highlighting its exceptional robustness.

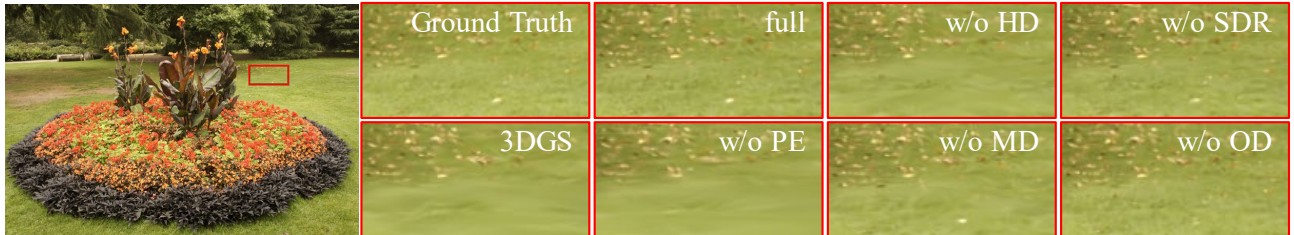

*Figure 15.* Visual results of the ablation study, highlighting the impact of each module on reconstruction quality.

*Table 11.* Ablation studies on hyperparameters, with the **adopted settings** highlighted. All metrics are evaluated on the Mip-NeRF 360 dataset and averaged across scenes.

| $H.P.$ | Value | PSNR↑ | SSIM↑ | LPIPS↓ | #G↓ |
|---|---|---|---|---|---|
| | **0.1** | 28.01 | 0.839 | 0.172 | 2.69M |
| $\lambda_S$ | 0.3 | 27.82 | 0.835 | 0.181 | 2.10M |
| | 0.5 | 27.48 | 0.823 | 0.196 | 1.92M |
| | 10 | 28.05 | 0.841 | 0.166 | 3.61M |
| $\tau_h^\omega$ | 15 | 28.00 | 0.840 | 0.169 | 3.09M |
| | **25** | 28.01 | 0.839 | 0.172 | 2.69M |
| | **10** | 28.01 | 0.839 | 0.172 | 2.69M |
| $\tau_m^\omega$ | 15 | 27.98 | 0.838 | 0.173 | 2.65M |
| | 25 | 27.97 | 0.838 | 0.174 | 2.63M |
| | **1000** | 28.01 | 0.839 | 0.172 | 2.69M |
| $Iter_h$ | 1500 | 27.95 | 0.838 | 0.174 | 2.57M |
| | 2000 | 27.93 | 0.837 | 0.175 | 2.52M |
| | 1000 | 27.92 | 0.839 | 0.172 | 2.70M |
| $Iter_m$ | **1500** | 28.01 | 0.839 | 0.172 | 2.69M |
| | 2000 | 27.98 | 0.839 | 0.173 | 2.66M |

## F. Per Scene Quantitative Result Integrating the Proposed Method with Existing Works

In this section, we provide per scene quantitative results on additional metrics to highlight the effectiveness of integrating our method with existing approaches. As shown in Table 24, Table 25, Table 26, Table 27, Table 28, Table 29, Table 30, Table 31, Table 32, Table 33, Table 34, Table 35, we integrate our method with 3DGS and Pixel-GS, achieving significant improvements in both reconstruction quality and efficiency. While the vanilla Pixel-GS demonstrates a poor quality-efficiency trade-off on BungeeNeRF, our method markedly enhances its performance in large-scale scenes, as detailed in Table 35. In Table 36, Table 37, Table 38, we integrate our method with CoR-GS. Since CoR-GS fails to distribute a sufficient number of Gaussian primitives for high-quality reconstruction under sparse-view settings, we only report the results in terms of quality metrics for comparison. Although the original method achieves slightly higher PSNR and SSIM in certain scenes due to blurriness, our method consistently outperforms CoR-GS in the perceptual metric LPIPS across all scenes, indicating superior perceptual quality.

*Table 12.* Per scene quantitative results on Mip-NeRF 360, Tanks & Temples and Deep Blending, comparing our method with state-of-the-art methods in terms of PSNR↑.

| Method | PSNR↑ | | | | | | | | | | | | |
|---|---|---|---|---|---|---|---|---|---|---|---|---|---|
| | Bicycle | Bonsai | Counter | Kitchen | Room | Stump | Garden | Flowers | Treehill | Train | Truck | Drjohnson | Playroom |
| 3DGS* | 25.617 | 32.349 | 29.144 | 31.450 | 31.628 | 26.913 | 27.735 | 21.808 | 22.736 | 21.768 | 25.452 | 29.139 | 29.935 |
| Pixel-GS* | 25.733 | 32.649 | 29.227 | 31.795 | 31.783 | 27.182 | 27.820 | 21.885 | 22.572 | 21.985 | 25.438 | 28.130 | 29.708 |
| Mini-Splatting-D | 25.55 | 31.72 | 28.72 | 31.75 | 31.41 | 27.11 | 27.67 | 21.50 | 22.13 | 21.04 | 25.43 | 29.32 | 30.43 |
| Taming-3DGS | 25.47 | 32.22 | 29.03 | 31.74 | 32.12 | 26.96 | 27.64 | 21.76 | 23.09 | 22.23 | 25.90 | 29.68 | 30.44 |
| Ours | 25.956 | 32.730 | 29.452 | 32.005 | 32.220 | 27.302 | 27.961 | 21.798 | 22.634 | 22.154 | 25.637 | 29.663 | 30.219 |

*Table 13.* Per scene quantitative results on Mip-NeRF 360, Tanks & Temples and Deep Blending, comparing our method with state-of-the-art methods in terms of SSIM↑.

| Method | SSIM↑ | | | | | | | | | | | | |
|---|---|---|---|---|---|---|---|---|---|---|---|---|---|
| | Bicycle | Bonsai | Counter | Kitchen | Room | Stump | Garden | Flowers | Treehill | Train | Truck | Drjohnson | Playroom |
| 3DGS* | 0.778 | 0.948 | 0.916 | 0.933 | 0.927 | 0.784 | 0.874 | 0.621 | 0.651 | 0.810 | 0.879 | 0.898 | 0.902 |
| Pixel-GS* | 0.792 | 0.951 | 0.920 | 0.936 | 0.930 | 0.797 | 0.878 | 0.652 | 0.652 | 0.823 | 0.883 | 0.886 | 0.900 |
| Mini-Splatting-D | 0.798 | 0.946 | 0.913 | 0.934 | 0.928 | 0.804 | 0.878 | 0.642 | 0.640 | 0.817 | 0.890 | 0.905 | 0.908 |
| Taming-3DGS | 0.78 | 0.94 | 0.91 | 0.93 | 0.92 | 0.78 | 0.87 | 0.61 | 0.65 | 0.81 | 0.89 | 0.91 | 0.91 |
| Ours | 0.805 | 0.953 | 0.922 | 0.936 | 0.936 | 0.807 | 0.877 | 0.654 | 0.657 | 0.826 | 0.888 | 0.905 | 0.908 |

*Table 14.* Per scene quantitative results on Mip-NeRF 360, Tanks & Temples and Deep Blending, comparing our method with state-of-the-art methods in terms of LPIPS↓.

| Method | LPIPS↓ | | | | | | | | | | | | |
|---|---|---|---|---|---|---|---|---|---|---|---|---|---|
| | Bicycle | Bonsai | Counter | Kitchen | Room | Stump | Garden | Flowers | Treehill | Train | Truck | Drjohnson | Playroom |
| 3DGS* | 0.205 | 0.173 | 0.178 | 0.113 | 0.191 | 0.208 | 0.103 | 0.329 | 0.319 | 0.209 | 0.147 | 0.247 | 0.246 |
| Pixel-GS* | 0.174 | 0.161 | 0.162 | 0.107 | 0.184 | 0.181 | 0.094 | 0.253 | 0.269 | 0.182 | 0.121 | 0.256 | 0.243 |
| Mini-Splatting-D | 0.158 | 0.175 | 0.172 | 0.114 | 0.190 | 0.169 | 0.090 | 0.255 | 0.262 | 0.181 | 0.100 | 0.218 | 0.204 |
| Taming-3DGS | 0.20 | 0.20 | 0.20 | 0.12 | 0.21 | 0.20 | 0.10 | 0.34 | 0.31 | 0.21 | 0.13 | 0.24 | 0.24 |
| Ours | 0.165 | 0.151 | 0.157 | 0.108 | 0.168 | 0.175 | 0.098 | 0.257 | 0.273 | 0.184 | 0.117 | 0.230 | 0.231 |

*Table 15.* Per scene quantitative results on Mip-NeRF 360, Tanks & Temples and Deep Blending, comparing our method with state-of-the-art methods in terms of the number of Gaussian primitives (#G)↓.

| Method | #G↓ | | | | | | | | | | | | |
|---|---|---|---|---|---|---|---|---|---|---|---|---|---|
| | Bicycle | Bonsai | Counter | Kitchen | Room | Stump | Garden | Flowers | Treehill | Train | Truck | Drjohnson | Playroom |
| 3DGS* | 5.78M | 1.25M | 1.17M | 1.75M | 1.49M | 4.73M | 5.07M | 3.38M | 3.62M | 1.08M | 2.58M | 3.28M | 2.33M |
| Pixel-GS* | 8.46M | 2.07M | 2.50M | 3.03M | 2.49M | 6.46M | 7.55M | 7.08M | 7.47M | 3.80M | 5.18M | 5.51M | 3.76M |
| Mini-Splatting-D | 6.03M | 3.78M | 3.75M | 3.78M | 4.05M | 5.41M | 5.81M | 4.87M | 4.86M | 3.95M | 4.58M | 4.91M | 4.35M |
| Taming-3DGS | 5.99M | 1.19M | 1.19M | 1.61M | 1.55M | 4.87M | 5.07M | 3.62M | 3.77M | 1.09M | 2.58M | 3.27M | 2.33M |
| Ours | 3.89M | 1.58M | 1.49M | 1.63M | 1.74M | 3.81M | 3.03M | 3.55M | 3.48M | 1.39M | 2.05M | 3.43M | 2.29M |

*Table 16.* Per scene quantitative results on Mip-NeRF 360, Tanks & Temples and Deep Blending, comparing our method with state-of-the-art methods in terms of rendering speed (FPS)↑.

| Method | FPS↑ | | | | | | | | | | | | |
|---|---|---|---|---|---|---|---|---|---|---|---|---|---|
| | Bicycle | Bonsai | Counter | Kitchen | Room | Stump | Garden | Flowers | Treehill | Train | Truck | Drjohnson | Playroom |
| 3DGS* | 100 | 310 | 244 | 195 | 235 | 151 | 122 | 200 | 176 | 285 | 208 | 160 | 228 |
| Pixel-GS* | 59 | 184 | 122 | 113 | 141 | 95 | 76 | 71 | 81 | 111 | 91 | 89 | 138 |
| Mini-Splatting-D | 107 | 135 | 109 | 112 | 137 | 121 | 105 | 127 | 125 | 117 | 113 | 147 | 170 |
| Taming-3DGS | 88 | 140 | 137 | 122 | 129 | 122 | 107 | 129 | 120 | 152 | 146 | 111 | 148 |
| Ours | 162 | 206 | 155 | 154 | 173 | 156 | 181 | 151 | 154 | 210 | 225 | 143 | 213 |

*Table 17.* Per scene quantitative results on Mip-NeRF 360, Tanks & Temples and Deep Blending, comparing our method with state-of-the-art methods in terms of the balance between quality and efficiency (QEB)↓.

| Method | QEB↓ | | | | | | | | | | | | |
|---|---|---|---|---|---|---|---|---|---|---|---|---|---|
| | Bicycle | Bonsai | Counter | Kitchen | Room | Stump | Garden | Flowers | Treehill | Train | Truck | Drjohnson | Playroom |
| 3DGS* | 1.185 | 0.070 | 0.085 | 0.101 | 0.121 | 0.652 | 0.428 | 0.556 | 0.656 | 0.079 | 0.182 | 0.506 | 0.251 |
| Pixel-GS* | 2.481 | 0.182 | 0.332 | 0.284 | 0.323 | 1.231 | 0.934 | 2.503 | 2.481 | 0.616 | 0.694 | 1.585 | 0.662 |
| Mini-Splatting-D | 0.890 | 0.490 | 0.592 | 0.385 | 0.562 | 0.756 | 0.498 | 0.978 | 1.019 | 0.827 | 0.644 | 0.728 | 0.522 |
| Taming-3DGS | 1.361 | 0.170 | 0.174 | 0.158 | 0.252 | 0.798 | 0.474 | 0.954 | 0.974 | 0.151 | 0.230 | 0.707 | 0.378 |
| Ours | 0.396 | 0.116 | 0.151 | 0.114 | 0.169 | 0.427 | 0.164 | 0.604 | 0.617 | 0.122 | 0.107 | 0.552 | 0.248 |

*Table 18.* Per scene quantitative results on BungeeNeRF, comparing our method with state-of-the-art methods in terms of PSNR↑.

| Method | PSNR↑ | | | | | | | |
|---|---|---|---|---|---|---|---|---|
| | Amsterdam | Barcelona | Bilbao | Chicago | Hollywood | Pompidou | Quebec | Rome |
| 3DGS* | 27.600 | 27.379 | 28.800 | 27.972 | 26.150 | 26.997 | 28.711 | 27.535 |
| Pixel-GS* | 27.756 | 26.780 | 28.582 | 27.524 | 26.121 | OOM | 28.245 | 26.583 |
| Mini-Splatting-D | 27.008 | 26.227 | 27.990 | 27.376 | 26.042 | 26.046 | 27.804 | 16.141 |
| Taming-3DGS | 27.553 | OOM | 28.849 | 28.292 | 26.408 | OOM | 28.900 | 27.484 |
| Ours | 27.887 | 27.546 | 29.081 | 28.390 | 26.146 | 27.180 | 29.013 | 27.647 |

*Table 19.* Per scene quantitative results on BungeeNeRF, comparing our method with state-of-the-art methods in terms of SSIM↑.

| Method | SSIM↑ | | | | | | | |
|---|---|---|---|---|---|---|---|---|
| | Amsterdam | Barcelona | Bilbao | Chicago | Hollywood | Pompidou | Quebec | Rome |
| 3DGS* | 0.913 | 0.915 | 0.915 | 0.927 | 0.868 | 0.916 | 0.931 | 0.914 |
| Pixel-GS* | 0.916 | 0.904 | 0.912 | 0.921 | 0.866 | OOM | 0.924 | 0.896 |
| Mini-Splatting-D | 0.909 | 0.894 | 0.911 | 0.920 | 0.865 | 0.901 | 0.920 | 0.568 |
| Taming-3DGS | 0.911 | OOM | 0.918 | 0.931 | 0.869 | OOM | 0.936 | 0.914 |
| Ours | 0.922 | 0.919 | 0.922 | 0.933 | 0.868 | 0.922 | 0.938 | 0.918 |

*Table 20.* Per scene quantitative results on BungeeNeRF, comparing our method with state-of-the-art methods in terms of LPIPS↓.

| Method | LPIPS↓ | | | | | | | |
|---|---|---|---|---|---|---|---|---|
| | Amsterdam | Barcelona | Bilbao | Chicago | Hollywood | Pompidou | Quebec | Rome |
| 3DGS* | 0.100 | 0.087 | 0.099 | 0.086 | 0.134 | 0.095 | 0.094 | 0.102 |
| Pixel-GS* | 0.095 | 0.101 | 0.103 | 0.090 | 0.138 | OOM | 0.106 | 0.129 |
| Mini-Splatting-D | 0.102 | 0.117 | 0.104 | 0.097 | 0.158 | 0.115 | 0.111 | 0.385 |
| Taming-3DGS | 0.113 | OOM | 0.100 | 0.088 | 0.150 | OOM | 0.096 | 0.112 |
| Ours | 0.087 | 0.084 | 0.092 | 0.081 | 0.140 | 0.089 | 0.087 | 0.098 |

*Table 21.* Per scene quantitative results on BungeeNeRF, comparing our method with state-of-the-art methods in terms of the number of Gaussian primitives(#G)↓.

| Method | #G↓ | | | | | | | |
|---|---|---|---|---|---|---|---|---|
| | Amsterdam | Barcelona | Bilbao | Chicago | Hollywood | Pompidou | Quebec | Rome |
| 3DGS* | 6.19M | 8.46M | 5.51M | 6.32M | 7.04M | 9.11M | 6.01M | 6.74M |
| Pixel-GS* | 10.26M | 11.04M | 7.98M | 9.76M | 9.88M | OOM | 8.37M | 8.08M |
| Mini-Splatting-D | 6.65M | 6.79M | 5.40M | 5.65M | 5.74M | 6.73M | 5.72M | 5.99M |
| Taming-3DGS | 6.20M | OOM | 5.51M | 6.30M | 7.06M | OOM | 6.04M | 6.75M |
| Ours | 4.96M | 5.97M | 3.76M | 4.48M | 4.88M | 6.12M | 4.57M | 5.03M |

*Table 22.* Per scene quantitative results on BungeeNeRF, comparing our method with state-of-the-art methods in terms of rendering speed(FPS)↑.

| Method | FPS↑ | | | | | | | |
|---|---|---|---|---|---|---|---|---|
| | Amsterdam | Barcelona | Bilbao | Chicago | Hollywood | Pompidou | Quebec | Rome |
| 3DGS* | 70 | 63 | 72 | 67 | 72 | 57 | 76 | 71 |
| Pixel-GS* | 42 | 48 | 54 | 43 | 52 | OOM | 57 | 62 |
| Mini-Splatting-D | 80 | 80 | 97 | 93 | 90 | 80 | 85 | 85 |
| Taming-3DGS | 63 | OOM | 66 | 70 | 65 | OOM | 66 | 66 |
| Ours | 85 | 83 | 88 | 92 | 97 | 78 | 98 | 87 |

*Table 23.* Per scene quantitative results on BungeeNeRF, comparing our method with state-of-the-art methods in terms of the balance between quality and efficiency(QEB)↓.

| Method | QEB↓ | | | | | | | |
|---|---|---|---|---|---|---|---|---|
| | Amsterdam | Barcelona | Bilbao | Chicago | Hollywood | Pompidou | Quebec | Rome |
| 3DGS* | 0.884 | 1.168 | 0.758 | 0.811 | 1.310 | 1.518 | 0.743 | 0.968 |
| Pixel-GS* | 2.321 | 2.323 | 1.522 | 2.043 | 2.622 | OOM | 1.557 | 1.681 |
| Mini-Splatting-D | 0.848 | 0.993 | 0.579 | 0.589 | 1.008 | 0.967 | 0.747 | 2.713 |
| Taming-3DGS | 1.112 | OOM | 0.835 | 0.792 | 1.629 | OOM | 0.879 | 1.145 |
| Ours | 0.508 | 0.604 | 0.393 | 0.394 | 0.704 | 0.698 | 0.406 | 0.567 |

*Table 24.* Per scene quantitative result of the proposed method is based on different models on Mip-NeRF 360, Tanks & Temples, and Deep Blending in terms of PSNR↑. Metrics are averaged across the scenes.

| Method | PSNR↑ | | | | | | | | | | | | |
|---|---|---|---|---|---|---|---|---|---|---|---|---|---|
| | Bicycle | Bonsai | Counter | Kitchen | Room | Stump | Garden | Flowers | Treehill | Train | Truck | Drjohnson | Playroom |
| 3DGS* | 25.617 | 32.349 | 29.144 | 31.450 | 31.628 | 26.913 | 27.735 | 21.808 | 22.736 | 21.768 | 25.452 | 29.139 | 29.935 |
| w/ Ours | 25.956 | 32.730 | 29.452 | 32.005 | 32.220 | 27.302 | 27.961 | 21.798 | 22.634 | 22.154 | 25.637 | 29.663 | 30.219 |
| Δ | +0.339 | +0.381 | +0.308 | +0.555 | +0.592 | +0.389 | +0.226 | -0.010 | -0.102 | +0.386 | +0.185 | +0.524 | +0.284 |
| Pixel-GS* | 25.733 | 32.649 | 29.227 | 31.795 | 31.783 | 27.182 | 27.820 | 21.885 | 22.572 | 21.985 | 25.438 | 28.130 | 29.708 |
| w/ Ours | 25.982 | 32.746 | 29.425 | 32.042 | 32.204 | 27.392 | 27.959 | 21.867 | 22.516 | 22.291 | 25.604 | 29.488 | 29.931 |
| Δ | +0.249 | +0.097 | +0.198 | +0.247 | +0.421 | +0.210 | +0.139 | -0.018 | -0.056 | +0.306 | +0.166 | +1.358 | +0.223 |

*Table 25.* Per scene quantitative result of the proposed method is based on different models on Mip-NeRF 360, Tanks & Temples, and Deep Blending in terms of SSIM↑.

| Method | SSIM↑ | | | | | | | | | | | | |
|---|---|---|---|---|---|---|---|---|---|---|---|---|---|
| | Bicycle | Bonsai | Counter | Kitchen | Room | Stump | Garden | Flowers | Treehill | Train | Truck | Drjohnson | Playroom |
| 3DGS* | 0.778 | 0.948 | 0.916 | 0.933 | 0.927 | 0.784 | 0.874 | 0.621 | 0.651 | 0.810 | 0.879 | 0.898 | 0.902 |
| w/ Ours | 0.805 | 0.953 | 0.922 | 0.936 | 0.936 | 0.807 | 0.877 | 0.654 | 0.657 | 0.826 | 0.888 | 0.905 | 0.908 |
| Δ | +0.027 | +0.005 | +0.006 | +0.003 | +0.009 | +0.023 | +0.003 | +0.033 | +0.006 | +0.016 | +0.009 | +0.007 | +0.006 |
| Pixel-GS* | 0.792 | 0.951 | 0.920 | 0.936 | 0.930 | 0.797 | 0.878 | 0.652 | 0.652 | 0.823 | 0.883 | 0.886 | 0.900 |
| w/ Ours | 0.809 | 0.953 | 0.922 | 0.936 | 0.936 | 0.812 | 0.880 | 0.663 | 0.657 | 0.832 | 0.885 | 0.901 | 0.900 |
| Δ | +0.017 | +0.002 | +0.002 | +0.000 | +0.006 | +0.015 | +0.002 | +0.011 | +0.005 | +0.009 | +0.002 | +0.015 | +0.000 |

*Table 26.* Per scene quantitative result of the proposed method is based on different models on Mip-NeRF 360, Tanks & Temples, and Deep Blending in terms of LPIPS↓.

| Method | LPIPS↓ | | | | | | | | | | | | |
|---|---|---|---|---|---|---|---|---|---|---|---|---|---|
| | Bicycle | Bonsai | Counter | Kitchen | Room | Stump | Garden | Flowers | Treehill | Train | Truck | Drjohnson | Playroom |
| 3DGS* | 0.205 | 0.173 | 0.178 | 0.113 | 0.191 | 0.208 | 0.103 | 0.329 | 0.319 | 0.209 | 0.147 | 0.247 | 0.246 |
| w/ Ours | 0.165 | 0.151 | 0.157 | 0.108 | 0.168 | 0.175 | 0.098 | 0.257 | 0.273 | 0.184 | 0.117 | 0.230 | 0.231 |
| Δ | -0.040 | -0.022 | -0.021 | -0.005 | -0.023 | -0.033 | -0.005 | -0.072 | -0.046 | -0.025 | -0.030 | -0.017 | -0.015 |
| Pixel-GS* | 0.174 | 0.161 | 0.162 | 0.107 | 0.184 | 0.181 | 0.094 | 0.253 | 0.269 | 0.182 | 0.121 | 0.256 | 0.243 |
| w/ Ours | 0.158 | 0.149 | 0.153 | 0.106 | 0.167 | 0.169 | 0.092 | 0.240 | 0.265 | 0.171 | 0.113 | 0.233 | 0.233 |
| Δ | -0.016 | -0.012 | -0.009 | -0.001 | -0.017 | -0.012 | -0.002 | -0.013 | -0.004 | -0.011 | -0.008 | -0.023 | -0.010 |

*Table 27.* Per scene quantitative result of the proposed method is based on different models on Mip-NeRF 360, Tanks & Temples, and Deep Blending in terms of the number of Gaussian primitives(#G)↓.

| Method | #G↓ | | | | | | | | | | | | |
|---|---|---|---|---|---|---|---|---|---|---|---|---|---|
| | Bicycle | Bonsai | Counter | Kitchen | Room | Stump | Garden | Flowers | Treehill | Train | Truck | Drjohnson | Playroom |
| 3DGS* | 5.78M | 1.25M | 1.17M | 1.75M | 1.49M | 4.73M | 5.07M | 3.38M | 3.62M | 1.08M | 2.58M | 3.28M | 2.33M |
| w/ Ours | 3.89M | 1.58M | 1.49M | 1.63M | 1.74M | 3.81M | 3.03M | 3.55M | 3.48M | 1.39M | 2.05M | 3.43M | 2.29M |
| Δ | -1.89M | +0.33M | +0.32M | -0.12M | +0.25M | -0.92M | -2.04M | +0.17M | -0.14M | +0.31M | -0.53M | +0.15M | -0.04M |
| Pixel-GS* | 8.46M | 2.07M | 2.50M | 3.03M | 2.49M | 6.46M | 7.55M | 7.08M | 7.47M | 3.80M | 5.18M | 5.51M | 3.76M |
| w/ Ours | 4.47M | 1.99M | 2.06M | 2.07M | 2.19M | 4.36M | 3.98M | 4.69M | 4.53M | 2.74M | 3.18M | 4.34M | 2.84M |
| Δ | -3.99M | -0.08M | -0.44M | -0.96M | -0.30M | -2.10M | -3.57M | -2.39M | -2.94M | -1.06M | -2.00M | -1.17M | -0.92M |

*Table 28.* Per scene quantitative result of the proposed method is based on different models on Mip-NeRF 360, Tanks & Temples, and Deep Blending in terms of rendering speed(FPS)↑.

| Method | FPS↑ | | | | | | | | | | | | |
|---|---|---|---|---|---|---|---|---|---|---|---|---|---|
| | Bicycle | Bonsai | Counter | Kitchen | Room | Stump | Garden | Flowers | Treehill | Train | Truck | Drjohnson | Playroom |
| 3DGS* | 100 | 310 | 244 | 195 | 235 | 151 | 122 | 200 | 176 | 285 | 208 | 160 | 228 |
| w/ Ours | 162 | 206 | 155 | 154 | 173 | 156 | 181 | 151 | 154 | 210 | 225 | 143 | 213 |
| Δ | +62 | -104 | -89 | -41 | -62 | +5 | +59 | -49 | -22 | -75 | +17 | -17 | -15 |
| Pixel-GS* | 59 | 184 | 122 | 113 | 141 | 95 | 76 | 71 | 81 | 111 | 91 | 89 | 138 |
| w/ Ours | 129 | 158 | 116 | 121 | 130 | 133 | 128 | 109 | 120 | 138 | 149 | 106 | 163 |
| Δ | +70 | -26 | -6 | +8 | -11 | +38 | +52 | +38 | +39 | +27 | +58 | +17 | +25 |

*Table 29.* Per scene quantitative result of the proposed method is based on different models on Mip-NeRF 360, Tanks & Temples, and Deep Blending in terms of the balance between quality and efficiency(QEB)↓.

| Method | QEB↓ | | | | | | | | | | | | |
|---|---|---|---|---|---|---|---|---|---|---|---|---|---|
| | Bicycle | Bonsai | Counter | Kitchen | Room | Stump | Garden | Flowers | Treehill | Train | Truck | Drjohnson | Playroom |
| 3DGS* | 1.185 | 0.070 | 0.085 | 0.101 | 0.121 | 0.652 | 0.428 | 0.556 | 0.656 | 0.079 | 0.182 | 0.506 | 0.251 |
| w/ Ours | 0.396 | 0.116 | 0.151 | 0.114 | 0.169 | 0.427 | 0.164 | 0.604 | 0.617 | 0.122 | 0.107 | 0.552 | 0.248 |
| Δ | -0.789 | +0.046 | +0.066 | +0.013 | +0.048 | -0.225 | -0.264 | +0.048 | -0.039 | +0.043 | -0.075 | +0.046 | -0.003 |
| Pixel-GS* | 2.481 | 0.182 | 0.332 | 0.284 | 0.323 | 1.231 | 0.934 | 2.503 | 2.481 | 0.616 | 0.694 | 1.585 | 0.662 |
| w/ Ours | 0.547 | 0.188 | 0.272 | 0.181 | 0.281 | 0.554 | 0.286 | 1.033 | 1.000 | 0.340 | 0.241 | 0.954 | 0.406 |
| Δ | -1.934 | +0.006 | -0.060 | -0.103 | -0.042 | -0.677 | -0.648 | -1.470 | -1.481 | -0.276 | -0.453 | -0.631 | -0.256 |

*Table 30.* Per scene quantitative result of the proposed method is based on different models on BungeeNeRF in terms of PSNR↑. Metrics are averaged across the scenes.

| Method | PSNR↑ | | | | | | | |
|---|---|---|---|---|---|---|---|---|
| | Amsterdam | Barcelona | Bilbao | Chicago | Hollywood | Pompidou | Quebec | Rome |
| 3DGS* | 27.600 | 27.379 | 28.800 | 27.972 | 26.150 | 26.997 | 28.711 | 27.535 |
| w/ Ours | 27.887 | 27.546 | 29.081 | 28.390 | 26.146 | 27.180 | 29.013 | 27.647 |
| Δ | +0.287 | +0.167 | +0.281 | +0.418 | -0.004 | +0.183 | +0.302 | +0.112 |
| Pixel-GS* | 27.756 | 26.780 | 28.582 | 27.524 | 26.121 | OOM | 28.245 | 26.583 |
| w/ Ours | 27.975 | 27.136 | 28.938 | 28.362 | 25.997 | 27.010 | 28.672 | 26.986 |
| Δ | +0.219 | +0.356 | +0.356 | +0.838 | -0.124 | — | +0.427 | +0.403 |

*Table 31.* Per scene quantitative result of the proposed method is based on different models on BungeeNeRF in terms of SSIM↑.

| Method | SSIM↑ | | | | | | | |
|---|---|---|---|---|---|---|---|---|
| | Amsterdam | Barcelona | Bilbao | Chicago | Hollywood | Pompidou | Quebec | Rome |
| 3DGS* | 0.913 | 0.915 | 0.915 | 0.927 | 0.868 | 0.916 | 0.931 | 0.914 |
| w/ Ours | 0.922 | 0.919 | 0.922 | 0.933 | 0.868 | 0.922 | 0.938 | 0.918 |
| Δ | +0.009 | +0.004 | +0.007 | +0.006 | +0.000 | +0.006 | +0.007 | +0.004 |
| Pixel-GS* | 0.916 | 0.904 | 0.912 | 0.921 | 0.866 | OOM | 0.924 | 0.896 |
| w/ Ours | 0.922 | 0.912 | 0.920 | 0.930 | 0.863 | 0.918 | 0.932 | 0.905 |
| Δ | +0.006 | +0.008 | +0.008 | +0.009 | -0.003 | — | +0.008 | +0.009 |

*Table 32.* Per scene quantitative result of the proposed method is based on different models on BungeeNeRF in terms of LPIPS↓.

| Method | LPIPS↓ | | | | | | | |
|---|---|---|---|---|---|---|---|---|
| | Amsterdam | Barcelona | Bilbao | Chicago | Hollywood | Pompidou | Quebec | Rome |
| 3DGS* | 0.100 | 0.087 | 0.099 | 0.086 | 0.134 | 0.095 | 0.094 | 0.102 |
| w/ Ours | 0.087 | 0.084 | 0.092 | 0.081 | 0.140 | 0.089 | 0.087 | 0.098 |
| Δ | -0.013 | -0.003 | -0.007 | -0.005 | +0.006 | -0.006 | -0.007 | -0.004 |
| Pixel-GS* | 0.095 | 0.101 | 0.103 | 0.090 | 0.138 | OOM | 0.106 | 0.129 |
| w/ Ours | 0.085 | 0.093 | 0.094 | 0.081 | 0.144 | 0.092 | 0.094 | 0.117 |
| Δ | -0.010 | -0.008 | -0.009 | -0.009 | +0.006 | — | -0.012 | -0.012 |

*Table 33.* Per scene quantitative result of the proposed method is based on different models on BungeeNeRF in terms of the number of Gaussian primitives(#G)↓.

| Method | #G↓ | | | | | | | |
|---|---|---|---|---|---|---|---|---|
| | Amsterdam | Barcelona | Bilbao | Chicago | Hollywood | Pompidou | Quebec | Rome |
| 3DGS* | 6.19M | 8.46M | 5.51M | 6.32M | 7.04M | 9.11M | 6.01M | 6.74M |
| w/ Ours | 4.96M | 5.97M | 3.76M | 4.48M | 4.88M | 6.12M | 4.57M | 5.03M |
| Δ | -1.23M | -2.49M | -1.75M | -1.84M | -2.16M | -2.99M | -1.44M | -1.71M |
| Pixel-GS* | 10.26M | 11.04M | 7.98M | 9.76M | 9.88M | OOM | 8.37M | 8.08M |
| w/ Ours | 6.60M | 6.64M | 4.57M | 5.58M | 5.85M | 7.39M | 5.46M | 5.27M |
| Δ | -3.66M | -4.40M | -3.41M | -4.18M | -4.03M | — | -2.91M | -2.81M |

*Table 34.* Per scene quantitative result of the proposed method is based on different models on BungeeNeRF in terms of rendering speed(FPS)↑.

| Method | FPS↑ | | | | | | | |
|---|---|---|---|---|---|---|---|---|
| | Amsterdam | Barcelona | Bilbao | Chicago | Hollywood | Pompidou | Quebec | Rome |
| 3DGS* | 70 | 63 | 72 | 67 | 72 | 57 | 76 | 71 |
| w/ Ours | 85 | 83 | 88 | 92 | 97 | 78 | 98 | 87 |
| Δ | +15 | +20 | +16 | +25 | +25 | +21 | +22 | +16 |
| Pixel-GS* | 42 | 48 | 54 | 43 | 52 | OOM | 57 | 62 |
| w/ Ours | 65 | 75 | 76 | 74 | 82 | 64 | 83 | 88 |
| Δ | +23 | +27 | +22 | +31 | +30 | — | +26 | +26 |

*Table 35.* Per scene quantitative result of the proposed method is based on different models on BungeeNeRF in terms of the balance between quality and efficiency(QEB)↓.

| Method | QEB↓ | | | | | | | |
|---|---|---|---|---|---|---|---|---|
| | Amsterdam | Barcelona | Bilbao | Chicago | Hollywood | Pompidou | Quebec | Rome |
| 3DGS* | 0.884 | 1.168 | 0.758 | 0.811 | 1.310 | 1.518 | 0.743 | 0.968 |
| w/ Ours | 0.508 | 0.604 | 0.393 | 0.394 | 0.704 | 0.698 | 0.406 | 0.567 |
| Δ | -0.376 | -0.564 | -0.365 | -0.417 | -0.606 | -0.820 | -0.337 | -0.401 |
| Pixel-GS* | 2.321 | 2.323 | 1.522 | 2.043 | 2.622 | OOM | 1.557 | 1.681 |
| w/ Ours | 0.863 | 0.823 | 0.565 | 0.611 | 1.027 | 1.062 | 0.618 | 0.701 |
| Δ | -1.458 | -1.500 | -0.957 | -1.432 | -1.595 | — | -0.939 | -0.980 |

*Table 36.* Per scene quantitative result of the proposed method is based on CoR-GS on 24-view Mip-NeRF 360 in terms of PSNR↑. Metrics are averaged across the scenes.

| Method | PSNR↑ | | | | | | |
|---|---|---|---|---|---|---|---|
| | Bicycle | Bonsai | Counter | Kitchen | Room | Stump | Garden |
| CoR-GS* | 20.496 | 24.905 | 23.331 | 21.973 | 25.376 | 19.819 | 19.941 |
| w/ Ours | 19.757 | 25.159 | 23.557 | 23.405 | 25.453 | 18.996 | 20.598 |
| Δ | -0.739 | +0.254 | +0.226 | +1.432 | +0.077 | -0.823 | +0.657 |

*Table 37.* Per scene quantitative result of the proposed method is based on CoR-GS on 24-view Mip-NeRF 360 in terms of SSIM↑.

| Method | SSIM↑ | | | | | | |
|---|---|---|---|---|---|---|---|
| | Bicycle | Bonsai | Counter | Kitchen | Room | Stump | Garden |
| CoR-GS* | 0.479 | 0.834 | 0.791 | 0.836 | 0.858 | 0.407 | 0.440 |
| w/ Ours | 0.465 | 0.847 | 0.797 | 0.855 | 0.854 | 0.399 | 0.549 |
| Δ | -0.014 | +0.013 | +0.006 | +0.019 | -0.004 | -0.008 | +0.109 |

*Table 38.* Per scene quantitative result of the proposed method is based on CoR-GS on 24-view Mip-NeRF 360 in terms of LPIPS↓.

| Method | LPIPS↓ | | | | | | |
|---|---|---|---|---|---|---|---|
| | Bicycle | Bonsai | Counter | Kitchen | Room | Stump | Garden |
| CoR-GS* | 0.491 | 0.212 | 0.213 | 0.158 | 0.177 | 0.605 | 0.529 |
| w/ Ours | 0.406 | 0.171 | 0.190 | 0.147 | 0.175 | 0.509 | 0.368 |
| Δ | -0.085 | -0.041 | -0.023 | -0.011 | -0.002 | -0.096 | -0.161 |

