# OpenReview forum: "Perceptual-GS: Scene-adaptive Perceptual Densification for Gaussian Splatting"
_ICML.cc/2025/Conference — ICML 2025 poster_

### Official Review · Reviewer_aPkn · 2025-03-09

**Overall Recommendation:** 3

**Summary:**

This paper focuses on improving the perceptual quality of 3DGS by 1) extracting edges to represent perceptual sensitivity and embedding it into the primitives for supervision; 2) introducing additional densification strategy based on the sensitivity. Experiments show that the proposed method achieves SOTA performance on multiple standard datasets while retaining efficiency.

**Claims And Evidence:**

The claims are supported in the evaluations. However, the transparency and interpretability can be improved.

**Essential References Not Discussed:**

Though the most related works about the specific track were involved, more 3DGS-related works with edge prior can be included for discussion like:

[1] Xiang, Haodong, et al. "Gaussianroom: Improving 3d gaussian splatting with sdf guidance and monocular cues for indoor scene reconstruction." arXiv preprint arXiv:2405.19671 (2024).

[2] Lin, Xin, et al. "HQGS: High-Quality Novel View Synthesis with Gaussian Splatting in Degraded Scenes." The Thirteenth International Conference on Learning Representations.

**Experimental Designs Or Analyses:**

Exhaustive experiments and ablation studies were designed to verify the effect of the contributions. However, the reported results mainly focus on the rendering quality, which causes a lack of interpretability. e.g., as in most 3DGS-related works, it's necessary to report more results about the depth, primitive distribution, and even surface normal if convenient, to enhance the transparency and interpretability of the paper. For this paper, the rendered sensitive map can also be visualized.

**Methods And Evaluation Criteria:**

The method and evaluation are technically suitable.

**Other Comments Or Suggestions:**

For suggestion, it's better to provide a demo video for such a 3D vision task to show the performance more intuitively and comprehensively.

**Other Strengths And Weaknesses:**

Strengths:

1. The proposed approaches of improving perceptual quality stem from a simple edge extraction, of which the conciseness may bring invaluable new insights to the community.
2. Detailed experiments and ablations are designed.
3. Experiment results show the method can achieve SOTA performance while keeping high efficiency. It can also adapt to large-scale scenes like BungeeNeRF.

Weaknesses:

1. The qualitative results somehow have a lack of interpretability. As in most 3DGS-related works, it's expected to report more qualitative
 results about the depth, primitive distribution, and even surface normal if convincing, to enhance the transparency and interoperability of the paper. For this paper, the rendered sensitive map can also be visualized.

2. As summarized in Table 8, many additional hyperparameters are introduced. This raises a concern about the robustness for different scenes, despite some ablation studies conducted in Table 9.

**Questions For Authors:**

See the weaknesses. May raise the rating if the concerns can be well solved.

## **Update after rebuttal**
The most recent reply addresses my concerns. I'll keep my rating to be positive.

**Relation To Broader Scientific Literature:**

None.

**Theoretical Claims:**

No theoretical claims needing proofs are involved in the paper.

---

> ### Author Rebuttal · Authors · 2025-04-01
>
> We thank the reviewer for the thoughtful response to our paper. We provide additional visualizations at https://akfwb.github.io/Perceptual-GS-Rebuttal/ and address specific points below:
>
> **Q1. Interpretable Visualizations**
>
> Thank you for your suggestion. Since our method is designed to optimize rendering quality rather than geometric structure, we do not include depth or normal maps in the original paper. However, based on your valuable feedback, we have provided additional visualizations in the supplementary link, including depth maps, primitive distributions, and rendered perceptual sensitivity maps. These visualizations will also be included in our revision.
>
> **Q2. Discussion about Works with Edge Prior**
>
> Although our method uses the Sobel operator, its goal is to identify perceptually sensitive regions, rather than extracting prominent edges. Therefore, we do not discuss 3DGS methods that rely on edge priors in the paper. Based on your suggestion, we will discuss and cite these works in the revision.
>
> **Q3. Concerns about the Thresholds and Hyperparameters**
>
> Our method does involve the use of certain thresholds and hyperparameters, which is a **common practice** among 3DGS-based approaches. Parameters such as loss weights and the interval of specific operations are essential for the proper execution of the method. However, unlike some prior works that require scene-specific manual tuning, our method adopts a **uniform set of hyperparameters across all experiments**, regardless of the dataset or the baseline it is combined with. This demonstrates the **generalizability and robustness** of our approach.
>
> In addition, some thresholds, such as the high-sensitivity scene threshold and the  scenes with sparse initial point cloud threshold, are determined based on **statistical analysis**. Due to space limitations, these implementation details are not included in the main paper, but they are available in the provided link. Based on your suggestion, we will include these details in the revision.
>
> Ablation studies and corresponding explanations, included in the Appendix, further demonstrate that the performance of our model is **not highly sensitive** to the specific values of these hyperparameters, indicating a reasonable degree of robustness. In the following tables, the selected values of each hyperparameter are highlighted in bold.
>
> |$\lambda_S$|PSNR|SSIM|LPIPS|#G|
> |-|-|-|-|-|
> |**0.1**|28.01|0.839|0.172|2.69M|
> |0.3|27.82|0.835|0.181|2.10M|
> |0.5|27.48|0.823|0.196|1.92M|
>
> |$\tau^{\omega}_h$|PSNR|SSIM|LPIPS|#G|
> |-|-|-|-|-|
> |10|28.05|0.841|0.166|3.61M|
> |15|28.00|0.840|0.169|3.09M|
> |**25**|28.01|0.839|0.172|2.69M|
>
> |$\tau^{\omega}_m$|PSNR|SSIM|LPIPS|#G|
> |-|-|-|-|-|
> |**10**|28.01|0.839|0.172|2.69M|
> |15|27.98|0.838|0.173|2.65M|
> |25|27.97|0.838|0.174|2.63M|
>
> |$Iter_h$|PSNR|SSIM|LPIPS|#G|
> |-|-|-|-|-|
> |**1000**|28.01|0.839|0.172|2.69M|
> |1500|27.95|0.838|0.174|2.57M|
> |2000|27.93|0.837|0.175|2.52M|
>
> |$Iter_m$|PSNR|SSIM|LPIPS|#G|
> |-|-|-|-|-|
> |1000|27.92|0.839|0.172|2.70M|
> |**1500**|28.01|0.839|0.172|2.69M|
> |2000|27.98|0.839|0.173|2.66M|
>
> **Q4. Demo Video for Better Visualization**
>
> Thank you for your suggestion. A demo video is available at the provided link for better visualization.

---

> > ### Comment · Reviewer_aPkn · 2025-04-03
> >
> > Thanks for the reply. However, the provided geometry results raised some additional concerns about the quality of reconstruction. As shown in the depth maps, there are many little noises near the camera, which is very like the small noisy primitives after the densification. And there are some brush-like traces in the depth map in textureless regions that means a wrong geometry. I'm concerned about the robustness of these regions when view changes. The shown video of bicycle is a relatively simple scene that can be well-reconstructed by many methods, which is less persuadable for the performance. Though the paper aims at the rendering quality, geometry is the core problem of the task, and the geometry results can reflect the real quality more faithfully than just the static RGB rendering or quantitative scores. I'm curious if these concerns can be well addressed.

---

> > > ### Author Response · Authors · 2025-04-04
> > >
> > > Thank you for your prompt feedback. We address the remaining concerns as follows:
> > >
> > > **Further Explanation of Geometric Reconstruction**
> > >
> > > We first address the issue regarding minor noise and brush-like traces observed in the depth maps. In Figure 6 at the provided link, the visualizations of depth maps are generated following the approach **used in Mini-Splatting**: for each pixel, only the Gaussian primitive **with the highest weight** is considered. This approach is chosen to illustrate how **the density of Gaussian primitives** varies across regions with different perceptual sensitivities. Under this strategy, the depth value of each pixel is determined by the most contributing primitive, causing the depth map to **appear discontinuous**, as the depth varies abruptly between neighboring pixels dominated by different primitives, and can lead to visual artifacts such as minor noise and brush-like traces in the rendered depth maps. While this method helps to visualize primitive distribution, we acknowledge that **it may have caused confusion** about the geometric reconstruction quality.
> > >
> > > To provide a more accurate assessment of our geometric reconstruction performance, we have re-rendered the depth maps using **the standard alpha-blending** method and compared our results against Pixel-GS. The updated results, which better reflect the geometric fidelity of our approach, are now available **in Figure 7 at the same provided link**, and **do not** exhibit the noise or brush-like traces present in the depth maps generated with the Mini-Splatting method. Due to the absence of explicit depth supervision, our method may underperform in depth rendering compared to approaches that incorporate depth constraints. However, compared to **other state-of-the-art methods** of the same type **without depth supervision**, our approach demonstrates **more accurate depth reconstruction** in certain regions.
> > >
> > > From a methodology perspective, optimizing 3DGS from the **image view** and from the **geometry view** are two common yet distinct directions. Perceptual-GS belongs to the former, where the distribution of primitives is optimized based on the perceptual characteristics of the 2D image space. The core motivation is to guide densification using **human visual sensitivity** to different image regions. Other works in this category include Pixel-GS and Mini-Splatting, which consider the 2D projected size of primitives.
> > >
> > > On the other hand, geometry-centric approaches often incorporate **additional geometric cues** during training, such as depth maps or normal maps. Examples include GSDF [r1] and GaussianRoom [r2]. We agree that geometric accuracy plays a crucial role in novel view synthesis tasks, and we consider this an important direction for our future research.
> > >
> > > [r1] Yu, Mulin, et al. "GSDF: 3DGS Meets SDF for Improved Rendering and Reconstruction."  In Proceedings of the Annual Conference on Neural Information Processing Systems, 2024.
> > >
> > > [r2] Xiang, Haodong, et al. "Gaussianroom: Improving 3D Gaussian Splatting with SDF Guidance and Monocular Cues for Indoor Scene Reconstruction." arXiv preprint arXiv:2405.19671 (2024).
> > >
> > > **More Demo Videos**
> > >
> > > The bicycle scene from the MipNeRF 360 dataset is a highly representative example that has been widely used in prior works [r1], [r2] to demonstrate better performance. To better illustrate the effectiveness of our method, we include comparison with the original 3DGS on this scene. Additionally, based on your suggestion, we include a demo video of **the flowers scene**, which contains more complex textures, for further visualization. Our method significantly reduces blurriness and produces results that are more consistent with **human visual perception**.
> > >
> > > [r1] Niedermayr, Simon, Josef Stumpfegger, and Rüdiger Westermann. "Compressed 3D Gaussian Splatting for Accelerated Novel View Synthesis." Proceedings of the IEEE/CVF Conference on Computer Vision and Pattern Recognition. 2024.
> > >
> > > [r2] Mallick, Saswat Subhajyoti, et al. "Taming 3DGS: High-quality Radiance Fields with Limited Resources." SIGGRAPH Asia 2024 Conference Papers. 2024.

---

### Official Review · Reviewer_wQ47 · 2025-03-11

**Overall Recommendation:** 3

**Summary:**

This paper proposes a 3D Gaussian Splatting (3DGS) optimization method that leverages additional perceptual sensitivity. By incorporating visual sensitivity, specifically edge response, the approach enables more fine-grained optimization of Gaussian representations. The perceptual sensitivity-adaptive densification strategy includes handling perceptually poor regions and scene-adaptive depth reinitialization. The experiments follow the existing pipeline.

**Claims And Evidence:**

This paper argues that incorporating perceptual sensitivity into 3DGS can lead to more effective optimization, enabling higher-quality rendering with a limited number of Gaussians. The approach is motivated by the observation that the adaptive density control in conventional 3DGS struggles to perform densification effectively in regions with complex geometry. To address this issue, perceptual sensitivity is explicitly optimized and used to guide the densification process. An ablation study demonstrates that removing the proposed perceptual densification results in a performance drop of approximately 0.3 dB in PSNR.

**Essential References Not Discussed:**

The opacity decline resembles the cloning function in Eq. 9 of 3DGS-MCMC. A comparison and discussion on this similarity are necessary.

[1] Kheradmand S., et al., "3D Gaussian Splatting as Markov Chain Monte Carlo", NeurIPS, 2024.

**Experimental Designs Or Analyses:**

Experiments are conducted on the Mip-NeRF 360, Deep Blending, Tanks & Temples, and BungeeNeRF datasets. The proposed method is compared with approaches that optimize 3D Gaussian Splatting under limited computational resources.

**Methods And Evaluation Criteria:**

This paper introduce four major modules.

1. **Perceptual Sensitivity Extraction**.
A binary map is generated using edge detection and smoothing. Thresholding is performed with a Sobel filter to extract perceptual sensitivity.

2. **Dual-Branch Rendering**.
A sensitivity parameter is introduced for each Gaussian. In simple regions, the number of Gaussians is constrained to improve efficiency. To account for multi-view consistency, the sensitivity term is learned, and during rendering, the sensitivity map is estimated through binary classification.

3. **Perceptual Sensitivity-Guided Densification**.
Densification is adapted based on perceptual sensitivity. The learned sensitivity values are used to determine gradient thresholding levels, ensuring more refined optimization.

4. **Scene-Adaptive Depth Reinitialization**.
Gaussian distributions are adjusted to improve rendering quality. Large Gaussians and those with high sensitivity undergo depth reinitialization using a mini-splatting approach.

The method is evaluated using rendering metrics such as PSNR, SSIM, and LPIPS, as well as the quality-efficiency balance (QEB) metric to assess trade-offs between rendering quality and computational efficiency.

**Other Comments Or Suggestions:**

No additional comments.

**Other Strengths And Weaknesses:**

There are concerns about novelty since parts of the proposed method overlap with existing approaches. The perceptual sensitivity extraction modifies the score function measure from Taming-3DGS and applies it to thresholding, while depth reinitialization follows the approach used in Mini-Splatting, making the method appear more like an engineered combination of existing techniques.

QEB does not seem to be an intuitive metric. While it aims to represent an overall trade-off, it produces unfavorable conclusions for models like EAGLES[2] and Efficient GS[3], where the number of Gaussians is significantly reduced compared to LPIPS. These models, characterized by a very low number of Gaussians and high FPS, yield a QEB score as low as 0.01, which fails to effectively capture the overall performance trade-off.

[2] Girish S., et al., "EAGLES: Efficient Accelerated 3D Gaussians with Lightweight EncodingS", ECCV, 2024.

[3] Lee, J. C., et al., "Compact 3D Gaussian Representation for Radiance Field", CVPR, 2024.

This paper introduces an adaptive densification strategy based on perceptual complexity, but it lacks a detailed analysis of the associated trade-offs. It is unclear how the distribution of Gaussians differs between texture-rich and the other areas, or how performance loss in regions where splitting does not occur effectively compares to the performance gain in perceptually complex areas.

**Questions For Authors:**

Perceptual sensitivity ultimately relies on the Sobel filter, raising questions about the actual performance gain in highly complex regions such as grass. Additionally, in Figure 2, the entire image appears masked, making it unclear how the proposed method behaves differently from standard 3DGS in such cases.

**Relation To Broader Scientific Literature:**

This paper introduces a method that adjusts gradient thresholding based on perceptual sensitivity to modify the optimization of Gaussians. It is related to approaches for optimizing Gaussians under limited computational resources and, more broadly, to methods for compressing the number of Gaussians.

**Theoretical Claims:**

No theoretical claims are made.

---

> ### Author Rebuttal · Authors · 2025-04-01
>
> We thank the reviewer for the thoughtful response to our paper. We provide additional visualizations at https://akfwb.github.io/Perceptual-GS-Rebuttal/ and address specific points below:
>
> **Q1. Discussion of 3DGS-MCMC**
>
> Eq. 9 in 3DGS-MCMC aims to **maintain** the opacity of a spatial region before and after cloning. However, as noted in our Appendix, cloned Gaussians are often poorly optimized. To avoid redundancy and mitigate negative effects during training, we apply **opacity decline** instead of preserving the original opacity as in 3DGS-MCMC. We will cite and discuss 3DGS-MCMC in the revision as suggested.
>
> **Q2. Claims about the Novelty**
>
> To the best of our knowledge, Perceptual-GS is the first approach to incorporate **explicit modeling of human visual perception** into 3D reconstruction, in addition to modeling color and geometry. While it shares certain similarities with prior works such as Taming-3DGS and Mini-Splatting, our method is fundamentally centered around **perceptual sensitivity**, leading to significant different design in both motivation and implementation.
>
> In Taming-3DGS, complex edges are extracted using a Laplacian filter, and edge complexity is determined by aggregating pixel response values over the projected areas of Gaussian primitives from multiple views. In contrast, our method derives a perceptually representative **sensitivity map**, **learns to** project 2D perceptual cues onto 3D primitives, and uses them to guide densification. Ablation studies (w/o PE) further demonstrate that using a binarized perceptual sensitivity map is more effective in identifying regions that require densification compared to traditional edge detection methods.
>
> |Method|PSNR|SSIM|LPIPS|#G|
> |-|-|-|-|-|
> |3DGS|27.71|0.826|0.202|3.14M|
> |w/o PE|27.74|0.825|0.204|2.09M|
> |Ours|28.01|0.839|0.172|2.69M|
>
> In Mini-Splatting, depth reinitialization is uniformly applied across all scenes. However, we observe that this strategy may have drawbacks in scenes that are already well reconstructed. To address this, we adaptively decide whether to apply it **based on the sensitivity learning process**. Notably, ablation results (w/o SDR) demonstrate that our method can still achieve significant improvements in reconstruction quality even without adaptive depth reinitialization.
>
> |Method|PSNR|SSIM|LPIPS|#G|
> |-|-|-|-|-|
> |3DGS|27.71|0.826|0.202|3.14M|
> |w/o SDR|27.93|0.832|0.176|2.68M|
> |Ours|28.01|0.839|0.172|2.69M|
>
> **Q3. The Effectiveness of QEB**
>
> Your concern is valid, as the QEB metric used in this paper primarily emphasizes efficiency. However, since both our method and the compared baselines aim to **improve the performance of 3DGS**, and given that the gains in #G and FPS are not as significant as those achieved by purely lightweight approaches, we believe this metric remains reasonable in this context. In addition to QEB, we also report #G, FPS, and LPIPS in the paper to provide a more comprehensive evaluation that accounts for both rendering efficiency and perceptual quality.
>
> **Q4. Differences Across Regions with Varying Perceptual Sensitivity**
>
> To verify that our approach does not compromise the quality of these regions, we apply the sensitivity maps to mask out sensitive areas in the final renderings on the MipNeRF 360 dataset and evaluate reconstruction quality on the remaining non-sensitive regions. The results confirm that our method **introduces no degradation** in these areas. Additional visualizations are available in the provided link.
>
> |  Method  |  PSNR  |  SSIM  |  LPIPS  |
> |----------|--------|--------|---------|
> |   3DGS   |  40.18 |  0.990 | 0.014   |
> |   Ours   |  40.72 |  0.991 | 0.014   |
>
> Based on your suggestion, we will include the experiment in our revision and provide additional visualizations to better illustrate how the perceptual sensitivity map guides the distribution of Gaussian primitives in different regions, making the underlying mechanism more interpretable.
>
> **Q5. The Effect of Sensitivity Map**
>
> Directly relying on the absolute response values of gradient maps derived from the Sobel operator can be misleading. Therefore, in our method, we enhance these maps by **simulating multiple characteristics of the human visual system (HVS)**, leading to significant quality improvements, as demonstrated by the results of the w/o PE ablation study presented in Q2.
>
> The sensitivity map reflects how the HVS perceives different regions. As shown in Fig. 3, smooth sky and complex grass areas exhibit distinct values, leading to different densification strategies. In Fig. 2, most pixel values are 1, indicating a high density of perceptually sensitive regions. Our method distributes more primitives in these areas, including those that **the original 3DGS fails to densify**, resulting in clear differences. More visual examples can be found at the provided link.

---

> > ### Comment · Reviewer_wQ47 · 2025-04-07
> >
> > I appreciate the authors' response. The provided results have been helpful in understanding the paper. However, I have several follow-up questions.
> >
> > The authors claim that they explicitly measure perceptual sensitivity, yet this paper does not propose a novel method for directly measuring perceptual sensitivity itself. Rather, it computes scores based on edge responses from images, subsequently optimizing the Gaussians according to these scores. While explicitly identifying the Gaussians to be optimized has some novelty, claiming that this process integrates perceptual sensitivity seems somewhat of an overstatement. Consequently, this model seems more close to an engineering approach aimed at lowering the threshold around edge regions rather than addressing general perceptual sensitivity. Therefore, a concern remains as to whether this aspect provides significant novelty compared to existing works.
> >
> > From my understanding, SDR reinitializes Gaussians that are large in scale and have high sensitivity. However, depth reinitialization is introduced to address biased or uneven Gaussian distributions. The proposed method selectively refines Gaussians in high sensitivity regions, but low sensitivity regions do not necessarily imply uneven distributions. Thus, the justification for sensitivity being a necessary criterion for depth reinitialization appears insufficient.
> >
> > In the case of QEB, it is not a generally applicable metric and is used only for very limited comparisons. This metric seems suitable solely for measuring changes within models sharing the same baseline. Thus, I do not consider it meaningful, as it cannot reliably represent the general performance of models.

---

> > > ### Author Response · Authors · 2025-04-08
> > >
> > > Thank you for your prompt feedback. We address the remaining concerns as follows:
> > >
> > > **Further Claims about The Novelty**
> > >
> > > We sincerely thank the reviewer for recognizing the novelty of our perceptual modeling strategy in selecting Gaussians for densification. To address your concerns, we further elaborate on the innovation of our method and its differences from existing approaches.
> > >
> > > The perceptual sensitivity map forms the foundation of our method. While it is derived from the Sobel operator, we enhance it through perception-oriented enhancement to **simulate the JND thresholding nature** of the HVS, and apply perception-oriented smoothing to **mimic findings from eye-tracking studies**. Due to space constraints, please refer to our response to reviewer kxHE's Q3. for more details.
> > >
> > > In image processing, numerous works have incorporated human perceptual properties by leveraging gradient maps to reflect spatially varying visual sensitivity. Building on this long-standing line of research in 2D perceptual modeling, we are the first to explicitly model human perceptual sensitivity in the context of 3D reconstruction. We would like to emphasize that **our novelty builds upon the well-studied 2D perceptual modeling works rather than overstated**.
> > >
> > > Existing works such as Taming-3DGS project each Gaussian onto multiple 2D views and aggregate pixel responses from edge detection within the covered areas, transferring 2D edge information to 3D primitives. However, **as discussed in Section 3.4**, this approach does not enforce consistent pixel responses across different views. For example, if a Gaussian projects to pixels with values of 1 in one view and 0 in another, it fails to capture consistent 2D information. In contrast, our **learning-based** method enforces multi-view consistency by adaptively adjusting Gaussian parameters during training, resulting in more accurate projection of 2D sensitivity into 3D space and more effective guidance for densification. **Rather than simply lowering edge detection thresholds**, our method incorporates multiple human perceptual characteristics. As shown in the following table, directly applying edge detection (w/o PE) yields no improvement, further validating the effectiveness of our approach.
> > >
> > > |Method|PSNR|SSIM|LPIPS|#G|
> > > |-|-|-|-|-|
> > > |3DGS|27.71|0.826|0.202|3.14M|
> > > |w/o PE|27.74|0.825|0.204|2.09M|
> > > |Ours|28.01|0.839|0.172|2.69M|
> > >
> > > In summary, our method is **well-motivated** and **not a simple engineering approach**. It innovatively introduces **human visual perception** into 3D reconstruction. Extensive results demonstrate its effectiveness: with only **72% of the parameters**, our method **outperforms** the original 3DGS in **perceptual quality** and achieves **a 30% speed-up** in large-scale scenes, while some SOTA methods encounter OOM or reconstruction failure. Moreover, our method is **generalizable** and can be applied to various 3DGS-based pipelines. More visualizations can be found at https://akfwb.github.io/Perceptual-GS-Rebuttal/.
> > >
> > >
> > > **Explanations of the Scene-adaptive Depth Reinitialization (SDR)**
> > >
> > > We would like to clarify that SDR **does not** reinitialize large high-sensitive primitives. Instead, it is adaptively applied **across different scenes** based on the characteristics of their initial point clouds.
> > >
> > > Specifically, **as noted in Section 3.6**, we apply depth reinitialization (DR) to scenes with overly sparse initial point clouds. Such sparsity leads to oversized Gaussians with **inaccurate spatial positions**, often covering regions of **mixed sensitivity**. These Gaussians are more likely to be densified, but new primitives placed in the same incorrect areas preserve the error and limit performance. In such cases, DR facilitates the redistribution of densified primitives toward more accurate regions. In contrast, for scenes with high-quality initial point clouds, applying DR may disrupt well-learned information and degrade performance since the sampling strategy.
> > >
> > > To identify scenes with overly sparse initial point clouds, we measure the proportion $\gamma$ of **large medium-sensitive Gaussians** after warm-up. A high ratio indicates **poor spatial priors** from the initial point cloud, making more corresponding Gaussians in such scenes more prone to be densified, which can hinder reconstruction quality. Thus, applying DR in such cases is both necessary and well justified.
> > >
> > > Due to space constraints, our explanation of this strategy is relatively brief, which may have caused confusion. Based on your suggestion, we will include a more detailed explanation of the motivation and implementation of SDR in our revision.
> > >
> > > **Explanation of the QEB Metric**
> > >
> > > In the paper, we provide standard evaluation metrics for both quality and efficiency (LPIPS, #G, and FPS), which are widely adopted in prior works and sufficiently demonstrate the improvements achieved by our method. Based on your suggestion, we will remove the QEB metric in the revision.

---

### Official Review · Reviewer_kxHE · 2025-03-13

**Overall Recommendation:** 3

**Summary:**

The paper introduces Perceptual-GS, a method to improve 3D Gaussian Splatting (3DGS) for novel view synthesis by integrating a perceptual-sensitivity mechanism during training. Concretely, the authors compute gradient-based sensitivity maps to model human perception of local structures, then employ a dual-branch rendering pipeline—one for RGB and one for “sensitivity”—to guide Gaussian densification. This aims to focus resources where the human eye would notice errors, thus achieving higher quality with fewer primitives. Further refinements (like scene-adaptive depth re-initialization) bolster performance in large-scale scenes. Experiments on Mip-NeRF 360, Tanks & Temples, Deep Blending, and BungeeNeRF show improvements in both quality (PSNR/SSIM/LPIPS) and efficiency (#Gaussians, FPS) over baselines.

**Claims And Evidence:**

Claim: The approach outperforms prior 3DGS variants in both perceptual quality and efficiency.

Evidence: The authors provide side-by-side comparisons showing modest gains in PSNR/SSIM and reduced LPIPS, often with fewer splats. They also include ablations that remove modules (e.g., perceptual densification) to demonstrate each part’s impact.

Concern: While the reported improvements are consistent, they remain incremental over existing 3DGS-based approaches. Gains in absolute image quality are often relatively small, underscoring the primarily engineering nature. Note that some state-of-the-art methods,eg, mip-splatting, scaffold-gs are not included for comparison.

**Essential References Not Discussed:**

Some older works on adaptive sampling or foveated rendering in computer graphics, as well as NeRF variants that use perceptual losses, might be relevant. Without them, the novelty claim is overstated—adaptive, perceptual-driven resource allocation is well-explored historically.

**Experimental Designs Or Analyses:**

Experiments are comprehensive and well-documented across multiple datasets, including large-scale scenes where some baselines fail. The improvements generally validate the approach but reinforce that this is an incremental engineering improvement. Although, as mentioned above,  some state-of-the-art methods,eg, mip-splatting, scaffold-gs are not included for comparison.

**Methods And Evaluation Criteria:**

The method is thoroughly tied to 3DGS: it modifies the distribution of Gaussians via a perceptual branch.
Evaluation focuses on standard reconstruction metrics (PSNR, SSIM, LPIPS) plus model size (number of Gaussians) and render speed.
Overall, the methodology is competent for a practical 3D rendering system. However, it offers no major departure in algorithmic or theoretical frameworks beyond applying known perceptual cues to 3DGS densification.

**Other Comments Or Suggestions:**

The paper may find a better fit at a vision/graphics venue, where incremental improvements in rendering quality/efficiency are more broadly appreciated.

**Other Strengths And Weaknesses:**

**Strengths**:

	1. The paper offers a clear quality-efficiency trade-off improvement.

	2. The paper has relatively thorough experiments, solid ablations, and stable results in large scenes.

	3. Potentially easy to integrate with other 3DGS-based pipelines.

**Weaknesses**:

	1. Limited novelty: mostly extends known densification heuristics with gradient-based sensitivity.

	2. Engineering-heavy with multiple thresholds and hyperparameters and does not propose a general ML contribution.

        3. Some state-of-the-art methods,eg, mip-splatting, scaffold-gs are not included for comparison.

**Questions For Authors:**

Please refers to the Strengths And Weaknesses part.

**Relation To Broader Scientific Literature:**

The paper is heavily rooted in 3DGS developments, citing many concurrent methods that tackle densification.
While it references fundamental perceptual concepts (e.g. gradient magnitude, SSIM), it does not connect deeply with prior research on user studies, advanced HVS modeling, or more general ML frameworks.
The method’s incremental nature aligns it more with specialized vision/graphics venues.

**Theoretical Claims:**

The paper provides no formal theory—it relies on empirical heuristics for densification. No new mathematical results; the approach is closer to a system-level enhancement than a theoretical innovation.

---

> ### Author Rebuttal · Authors · 2025-04-01
>
> We thank the reviewer for the thoughtful response to our paper. We provide additional visualizations at https://akfwb.github.io/Perceptual-GS-Rebuttal/ and address specific points below:
>
> **Q1. Concerns about Engineering-driven Incremental Improvements**
>
> To the best of our knowledge, our work is the first to introduce **explicit human perceptual modeling** into the field of 3D reconstruction, aiming to generate perceptually aligned 3D scenes. The proposed method is simple yet effective, with a clear motivation. Comparing with existing approaches, it shows less blurriness and artifacts, significantly improves perceptual quality. More visualizations can be found in our provided link. Besides, our method uses **less than 60%** of the parameters while **surpassing** the reconstruction quality of other quality-focused approaches, resulting in a significant improvement in overall efficiency. Moreover, our method is **general** and can be seamlessly integrated with various 3DGS-based approaches—such as Pixel-GS and CoR-GS—to further enhance reconstruction quality even under sparse-view settings. The quantitative results are as follows:
>
> |Method|PSNR|SSIM|LPIPS|
> |-|-|-|-|
> |Pixel-GS|27.85|0.834|0.176|
> |w/ Ours|28.01|0.841|0.167|
>
> |Method|PSNR|SSIM|LPIPS|
> |-|-|-|-|
> |CoR-GS|22.26|0.664|0.341|
> |w/ Ours|22.42|0.681|0.281|
>
> **Q2. Concerns about Novelty**
>
> While previous works on adaptive sampling, foveated rendering and NeRF variants with perceptual losses are indeed related to the general concept of human perception, Perceptual-GS is the first to **explicitly model the human visual perception** within the 3D reconstruction process. Our method jointly incorporates geometric, color, and perceptual sensitivity models to represent the scene, leading to reconstructions that better align with visual perception. Following your suggestion, we will cite and include a more detailed discussion of these related works in the revision to better highlight our novelty.
>
> **Q3. Connections to Prior Perception-Related Works**
>
> The concept of visual perception has been extensively studied in the field of image processing. However, in the domain of 3D reconstruction, although some prior works have explored perceptual concepts, they often rely on perceptual loss **without explicitly modeling** the properties of the HVS. Others employ foveated rendering to improve efficiency, but fail to achieve an overall improvement in quality, as the references *Lin et al., 2024* and *Franke et al., 2024* cited in the Introduction of our paper. In contrast, our approach introduces an **explicit perceptual representation** and models multiple key characteristics of the HVS:
> * The HVS is **highly sensitive to regions with rich local structures** [r1]. To capture this characteristic, we employ the Sobel operator to extract gradients, which serve as an effective representation of HVS sensitivity across different regions of the image.
> * Due to **the thresholding nature** of human perception characterized by the Just Noticeable Difference [r2], we enhance the gradient magnitude maps through Perception-oriented Enhancement, effectively suppress the misleading influence of absolute gradient values.
> * **Eye-tracking studies** have shown that adjacent gaze points are often merged into a single fixation, typically representing a region with remarkable difference relative to its surroundings [r3]. To simulate this, we apply Perception-oriented Smoothing to generate a final binary perceptual sensitivity map that is both aligned with human perception and easy to learn.
>
> Based on your suggestion, we will emphasize the connection between the proposed method and key properties of the HVS in our revision.
>
> [r1] Xue, Wufeng, et al. "Gradient Magnitude Similarity Deviation: A Highly Efficient Perceptual Image Quality Index." IEEE Transactions on Image Processing 23.2 (2013): 684-695.
>
> [r2] Lubin, Jeffrey. "A Human Vision System Model for Objective Picture Quality Measurements." 1997 International Broadcasting Convention IBS 97. IET, 1997.
>
> [r3] Gu, Ke, et al. "Saliency-guided Quality Assessment of Screen Content Images." IEEE Transactions on Multimedia 18.6 (2016): 1098-1110.
>
> **Q4. Comparison with More SOTAs**
>
> As our method targets a different problem, we do not compare it with efficiency-focused Scaffold-GS or quality-focused Mip-Splatting in the paper. Based on your suggestion, we now include comparisons with them. Since Scaffold-GS does not explicitly store all primitives, its #G metric is omitted.
>
> |Method|PSNR|SSIM|LPIPS|FPS|#G|
> |-|-|-|-|-|-|
> |Scaffold-GS|27.99|0.825|0.207|476|-|
> |Mip-Splatting|27.79|0.827|0.203|125|3.97M|
> |Ours|28.01|0.839|0.172|166|2.69M|
>
> **Q5. Concerns about the Thresholds and Hyperparameters**
>
> Due to space constraints, we refer the reviewer to our response to reviewer aPkn's Q3., which addresses the same issue in detail.
>
> **Q6. Issues about the Supplementary Material**
>
> We have provided detailed Appendix after the References.

---

> > ### Comment · Reviewer_kxHE · 2025-04-07
> >
> > I appreciate the authors' detailed response. First of all, I must say sorry to the authors about neglecting the provided supplementary materials. Based on the rebuttal, which address most of my concerns, especially for the experiment parts, and the feedback from other reviewers, I am happy to keep my rating to be positive.

---

> > > ### Author Response · Authors · 2025-04-07
> > >
> > > Thank you very much for your thoughtful comments and appreciation of our work. We will do our best to improve the final version of our paper based on your valuable suggestions.

---

### Official Review · Reviewer_9bnu · 2025-03-13

**Overall Recommendation:** 4

**Summary:**

Perceptual-GS addresses a core limitation of 3D Gaussian Splatting (3DGS) for novel view synthesis by adaptively distributing Gaussian primitives based on human perceptual sensitivity. Traditional 3DGS methods suffer from either insufficient coverage in visually important areas or over-densification in simpler regions. Perceptual-GS tackles this by first modeling perceptual sensitivity from multi-view images, emphasizing scene details that the human eye is most sensitive to. It employs a dual-branch rendering framework, fusing both RGB and sensitivity signals to adaptively refine and distribute Gaussians, thereby boosting reconstruction quality while constraining the total number of primitives. Additionally, a scene-adaptive depth reinitialization mechanism further improves quality in regions with sparse initial geometry.

**Claims And Evidence:**

Perceptual-GS claims it can be effectively integrated with other 3DGS-based methods, while all of the experiments presented—including those in the supplementary materials—only include the proposed Perceptual module combined with 3DGS and Pixel-GS. It will be helpful if can show additional experiments to incorporate with other GS methods, which can prove its generalization, especially for sparse view GS methods, where the Gaussian distribution problem tends to be more pronounced.

While the paper provides a thorough quantitative evaluation, the limited visualizations shown appear to indicate only small improvements. Also as the paper claimed that the proposed method prioritizing the densification of Gaussian primitives in high-sensitivity regions to human perception and constraining their generation in low-sensitivity areas, it remains unclear whether this approach might compromise clarity or reduce quality in other parts of the scene. Therefore, it would be very helpful to include additional visualizations—either in the main paper or the supplementary material—to more clearly illustrate the advantages and potential trade-offs.

Perceptual Sensitivity map is unclear should looks like Fig 2 Sensitivity GT or should looks like Fig3 Perception-oriented Smoothing results, but both seems not very helpful for grass and ground as shown in Fig5.

**Essential References Not Discussed:**

No

**Experimental Designs Or Analyses:**

Please refer Claims And Evidence.

**Methods And Evaluation Criteria:**

Proposed method is make sense for the problem.

**Other Comments Or Suggestions:**

No

**Other Strengths And Weaknesses:**

The cropped region obscures the full image too much.

**Questions For Authors:**

Please refer Claims And Evidence.

**Relation To Broader Scientific Literature:**

Unknown

**Theoretical Claims:**

The paper not contain theoretical claim except some describtion about 3D Gaussian Splatting and Perceptual Sensitivity, which is correct.

---

> ### Author Rebuttal · Authors · 2025-04-01
>
> We thank the reviewer for the thoughtful response to our paper. We provide additional visualizations at https://akfwb.github.io/Perceptual-GS-Rebuttal/ and address specific points below:
>
> **Q1. Concerns about the Generalizability**
>
> Thank you for the valuable suggestion. To further demonstrate the generalizability of our approach, especially for sparse-view 3D Gaussian Splatting methods, we simply integrate our perceptual sensitivity-guided densification strategy with CoR-GS [r1] to validate its adaptability under sparse-view settings. Since the paper only provides code without the initial point cloud used for training, we followed the provided instructions to construct a training dataset containing only 24 views. We then trained both the CoR-GS and the version enhanced with our method under the same settings.
>
> |Method|PSNR|SSIM|LPIPS|
> |-|-|-|-|
> |CoR-GS|22.26|0.664|0.341|
> |w/ Ours|22.42|0.681|0.281|
>
> As shown in the table, our method achieves notable improvements even under sparse-view settings, particularly in the perceptual metric LPIPS. These results further validate the versatility and effectiveness of our approach. Qualitative comparisons are available in the link we provided.
>
> [r1] Zhang, Jiawei, et al. "CoR-GS: Sparse-view 3D Gaussian Splatting via Co-regularization." European Conference on Computer Vision, 2024.
>
> **Q2. Limited Visualization and Small Improvements**
>
> Due to space limitations in the main text, we have included additional visual results in the Appendix. Compared to other related methods, our approach achieves better reconstruction quality, particularly in challenging regions, **effectively reducing issues such as excessive blur and visual artifacts**. More qualitative comparisons are available in the provided link. Following your suggestion, we will incorporate more visual results in our revision.
>
> **Q3. Concerns about the Compromised Quality in Non-sensitive Areas**
>
> Since our method primarily targets perceptually sensitive regions, we do not include comparisons on non-sensitive areas in the experiments. To verify that our approach does not compromise the quality of these regions, we apply the sensitivity maps to mask out sensitive areas in the final renderings on the MipNeRF 360 dataset and evaluate reconstruction quality on the remaining non-sensitive regions. The results confirm that our method **introduces no degradation** in these areas. Additional visualizations are available in the provided link.
>
> |  Method  |  PSNR  |  SSIM  |  LPIPS  |
> |----------|--------|--------|---------|
> |   3DGS   |  40.18 |  0.990 | 0.014   |
> |   Ours   |  40.72 |  0.991 | 0.014   |
>
> Based on your suggestion, we will include the experiment in our revision.
>
> **Q4. Visualization of the Sensitivity Map**
>
> Figures 2 and 3 both visualize the perceptual sensitivity maps. Fig. 2 presents the full image, while Fig. 3 displays a cropped region to better highlight the variations in sensitivity across different areas. Compared to the baseline, our method distributes more Gaussian primitives in perceptually complex regions such as textured grass areas guided by the sensitivity map, resulting in images with higher visual quality as perceived by the human eye. This also helps to reduce noticeable blur and artifacts in the rendered results. Additional sensitivity map visualizations are available in the link we provided. Based on your suggestion, we will include additional visualizations of the sensitivity maps for further illustration in our revision.
>
> **Q5. The Cropped Region Obscures the Full Image Too Much**
>
> Thank you for your suggestion. We will revise the cropped region in the visualization accordingly in our revision.

---

> > ### Comment · Reviewer_9bnu · 2025-04-04
> >
> > Thanks for authors' response, they solve my questions, I change my overall recommendation to 4: Accept.

---

> > > ### Author Response · Authors · 2025-04-04
> > >
> > > Thank you very much for your thoughtful comments and appreciation of our work. We will do our best to
> > >  improve the final version of our paper based on your valuable suggestions.

---

### Decision · Program_Chairs · 2025-05-01

**Decision:**

Accept (poster)

**Comment:**

This paper proposes Perceptual-GS, a 3D Gaussian Splatting method guided by human perceptual sensitivity to improve rendering quality and efficiency. Reviewers appreciated the solid experiments and practical value, though some noted limited novelty and interpretability. The authors’ detailed rebuttal addressed most concerns, and all reviewers gave positive recommendations.